# ZooProbe: A Data Engine for Evaluating, Exploring, and Evolving Large-scale Training Data for Multimodal LLMs

**Yi-Kai Zhang**[1,2,3*]   **Shiyin Lu**[3]   **Qing-Guo Chen**[3]   **De-Chuan Zhan**[1,2]   **Han-Jia Ye**[1,2†]

[1]School of Artificial Intelligence, Nanjing University
[2]National Key Laboratory for Novel Software Technology, Nanjing University
[3]Alibaba International Digital Commerce

## ABSTRACT

Multimodal Large Language Models (MLLMs) are thriving through continuous fine-tuning by LLMs. Driven by the law that "scale is everything", MLLMs expand their training sets during version iterations. In this paper, we propose a large-scale training data engine built around an evaluating-exploring-evolving (E3) loop. Evaluating the data provides insights into its characteristics. Exploring quality rules helps identify which data enhances training. Together, these processes facilitate the systematic evolution of new, high-quality data. With the E3 loop, we introduce ZooProbe, an efficient data engine for MLLMs. First, the problem of data expansion is formalized as a tree of sampling and growth. ZooProbe introduces a small-scale model *zoo* to obtain comprehensive evaluations for child datasets. From multiple perspectives, visual, textual, and multimodal models cover over 50 dimensions of intrinsic and meta attributes, such as object and topic distribution, and higher-level properties, like annotation quality and scene complexity. ZooProbe constructs based on $A^*$ search, modeling the heuristic function as a quality estimate from data evaluation results. It dynamically explores the rule of data quality based on the model state of the *probe* datasets. Additionally, it evolves new targeted data with identified high-quality rules. We also develop an extra heuristic quality ranker with the data utilized and discarded during the expansion. Our experiments show that ZooProbe significantly breaks the scaling law in multimodal instruction fine-tuning at scales of $260k$ and below. ZooProbe generates high-quality data that accelerates MLLM training and enhances performance, automating the evolution of large-scale training data.

## 1 INTRODUCTION

The rapid advancements in Large Language Models (LLMs) (Du et al., 2022; Chiang et al., 2023; Jiang et al., 2023; OpenAI, 2022; Touvron et al., 2023a;b) have given rise to Multimodal LLMs (MLLMs), which build upon the foundations of LLMs and are continuously trained to process visual instructions (Chen et al., 2023; Zhu et al., 2024; Liu et al., 2023; 2024; Yao et al., 2024; OpenAI, 2023; 2024a;b). This growth often aligns with the "scale is everything" principle (Kaplan et al., 2020; Hoffmann et al., 2022), suggesting that generalization ability may follow a power-law relationship with the size of training data. Ongoing efforts focus on expanding visual fine-tuning datasets into developing the next generation of MLLMs (Bai et al., 2023; Wang et al., 2024).

Rich multimodal information in training data, especially the knowledge hierarchy and interaction between visual prompts and text, significantly boosts learning efficiency and performance. The data quality – reflecting the "garbage in, garbage out" – greatly influences learning speed and the depth of acquired knowledge (Power et al., 2022; Xie et al., 2023b; Lee et al., 2024; McKinzie et al., 2024; Li et al., 2024c). In the context of MLLM training data, researchers frequently introduce supplementary MLLM to label answers and even to generate visual instructions from images freely (Chen et al.,

---

*Work done during the internship at Alibaba International Digital Commerce.
†Corresponding author, email: yehj@lamda.nju.edu.cn.

2024a; Adler et al., 2024). Such a pipeline can limit knowledge guidance and result in a biased training set that relies on the preferences of data-generating models, such as most are captioning tasks with shallow understanding. Moreover, the multimodal QAs are significantly large as the Cambrian (Tong et al., 2024) consists of $10m$ entries. Therefore, effectively sampling from diverse existing data and generating new, targeted data is essential for developing next-generation datasets.

Firstly, a thorough evaluation can reveal the characteristics of the new training data and eliminate poor-quality entries. Existing data evaluation methods often focus on natural characteristics, such as text length and perplexity (Cao et al., 2023; Xu et al., 2024), or rely on fixed and uniform difficulty metrics from the additional scorer (Lian et al., 2023; Chen et al., 2024c). After obtaining the evaluation results of the data, it is crucial to explore a rule to determine what distribution, when added to the existing dataset, can contribute to forming a high-quality training set.

One of the high-quality selection rule expressions is the distribution ratio of dimensions (Wen et al., 2023; Fan et al., 2024; Lu et al., 2024a; Tong et al., 2024). Previous methods (Xue et al., 2021; Rae et al., 2021) relied on heuristic functions for threshold control, often using random or manually set values. In real-world applications, the definition of data quality changes with existing distributions and the model status on the available dataset (Zhou et al., 2023; Chen et al., 2024d). In addition to sampling from real-world sources, we can generate and evolve visual instructions based on identified rules. This data pipeline yields more efficient outputs, enhancing MLLMs to discover visual information, particularly regarding spatial, temporal, and deeper contextual reasoning.

These key elements construct a cyclical engine of comprehensive data evaluation, exploratory rule discovery, and targeted iterative evolution. Such an efficient data engine can fundamentally lower computing expenses and minimize the carbon footprint. In this paper, we introduce ZOOPROBE, a data engine for MLLM training that implements an evaluating-exploring-evolving (E3) loop. ZOOPROBE utilizes a small-scale model zoo to *evaluate* data, *explore* for high-quality data rules, and heuristically expand the training dataset. It also continuously *evolves* new data through rule-based guidance, constructing a seamless cycle from assessment to search to generation. ZOOPROBE produces high-quality data for MLLM version iterations, beating the limitations of scaling laws. Initially, ZOOPROBE derives data description vectors from a compact collection of visual, textual, and multimodal models, tackling over 50 *intrinsic* and *meta* dimensions. The *intrinsic* factors include attributes such as resolution, language, instruction length, visual objects, and topics. On the other hand, the *meta* factors are examined from a higher perspective, including annotation reliability, contextual relevance, external knowledge dependency, and multimodal interactivity. These dimensions offer valuable insights into the dataset's nature and properties without incurring feature preservation costs or requiring excessive backward computation (Gao et al., 2021).

ZOOPROBE then develops an $A^\star$-based search framework that includes the reward-so-far function and the future-training-quality estimates. Starting from a dynamic definition of high-data-quality based on model states, ZOOPROBE explores the optimal distribution through perturbations of the initial diverse one. It then establishes a connection between data descriptions and training quality, iterating to estimate quality scores of the data expansion. Besides sampling labeled visual instructions from diverse wild collections, ZOOPROBE integrates the identified high-quality rules and representative examples into an additional MLLM, enabling valuable data's efficient generation and evolution. Based on the proposed E3 loop, ZOOPROBE eliminates the cost of manual prompt engineering. Its meta-description vectors offer inherent interpretability, facilitating the discovery of new dimensions in scaling laws. Moreover, we specifically learn an MLLM to assess the data quality and validate its effectiveness. The experimental results show that ZOOPROBE can surpass the MLLM scaling law from $10k$ to $260k$. In summary, our contributions are as follows:

- We develop the E3 loop of the data engine using a heuristic search framework to expand datasets, evaluate diverse data dimensions, and adaptively explore quality while evolving new data.

- We present ZOOPROBE, which features a diverse model zoo for a thorough evaluation, establishes a probe set for targeted rule selection, and facilitates the evolution of missing distribution.

- During ZOOPROBE's expansion, we develop MLLM-Scorer to assess data quality. Experiments demonstrate that ZOOPROBE significantly beats scaling laws across scales from $10k$ to $260k$.

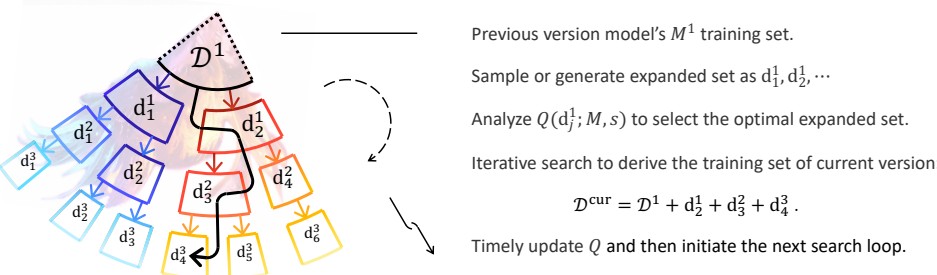

Previous version model's $M^1$ training set.

Sample or generate expanded set as $d_1^1, d_2^1, \cdots$

Analyze $Q(d_j^1; M, s)$ to select the optimal expanded set.

Iterative search to derive the training set of current version:

$$\mathcal{D}^{\text{cur}} = \mathcal{D}^1 + d_2^1 + d_3^2 + d_4^3 .$$

Timely update $Q$ and then initiate the next search loop.

Figure 1: **Illustration of dataset expansion tree.** As described in section 2, based on the current dataset, we perform a heuristic search for each expansion from multiple candidate batches, calculating the $Q$ values to select the optimal one. The solid black line represents the expansion path.

## 2 PRELIMINARY: TRAINING DATA EXPANSION IN MLLM ITERATIONS

The training of MLLMs involves iterative enhancements, and the critical step is expanding the training set. Starting with a base pre-trained MLLM $M^0$, the current version, denoted as $M^1$, is initialized from the $M^0$ and tuned on the instruction dataset $\mathcal{D}^1$. For the next version $M^2$, the dataset is expanded to $\mathcal{D}^2$. While it is expected to fine-tune $M^0$ on $\mathcal{D}^2$ and obtain the $M^2$, incrementally fine-tuning $M^1$ on $\mathcal{D}^2$ may risk catastrophic forgetting (Zhou et al., 2024; Zhang et al., 2024).

Consider the cumulative expansion process, *e.g.*, when $10k$ new instruction data is added, denoted as $\mathbf{d}^1$, we define dataset expansion as $\mathcal{D}^2$ being $\mathcal{D}^1$ plus $\mathbf{d}^1$. Thus, $\mathcal{D}^{\text{cur}} = \mathcal{D}^1 + \mathbf{d}^1 + \mathbf{d}^2 + \mathbf{d}^3 + \cdots$. In summary, our goal is to fine-tune $M^0$ on $\mathcal{D}^{\text{cur}}$ to achieve the optimal performance, where $\mathcal{D}^{\text{cur}}$ is expanded from the initial $\mathcal{D}^1$. In the context of expanding our dataset at each step, we consider the exploration efficiency and sample the next one from $n$ potential candidate batches, denoted as $\mathbf{d}_1^1, \mathbf{d}_2^1, \ldots, \mathbf{d}_n^1$. The most valuable batch (selected) improves the overall quality of the existing training set. These candidate batches are derived from an existing labeled instruction dataset $\mathcal{D}_{\text{labeled}}$, or be generated from unlabeled visual cues or other in-the-wild dataset $\mathcal{D}_{\text{unlabeled}}$.

As illustrated in Figure 1, the expansion process from $\mathcal{D}^i$ to $\mathcal{D}^{i+1}$ can be visualized as a growing tree of the dataset. Acquiring high-quality training data can thus be framed as the search for the optimal path. We introduce a heuristic strategy, where the tree structure at step $i$ generates $n$ candidate batches: $\mathbf{d}_1^i, \mathbf{d}_2^i, \ldots, \mathbf{d}_n^i$. Motivated by the $A^\star$ search framework, we propose a quality function:

$$Q\left(\mathbf{d}_j; M, s\right) = r\left(\mathbf{d}_j; M\right) + \gamma\, h\left(\mathbf{d}_j; \boldsymbol{\pi}^\star\left(s\right)\right) . \tag{1}$$

For simplicity, we omit the step-index $i$. Here, $M$ and $s$ represent the training model and its current state, respectively. $r(\cdot)$ is the reward-so-far function that indicates the quality score of adding $\mathbf{d}_j$ to the existing dataset; $\boldsymbol{\pi}^\star(s)$ is a rule for high-quality data based on the model state $s$, and $h(\,\cdot\,;\,\cdot\,)$ is a heuristic estimate of the future training quality of $\mathbf{d}_j$. The ratio $\gamma$ is the discount factor.

In Equation 1, it is adequate to calculate the component $r(\cdot)$, as it represents the existing performance score or training cost. Meanwhile, the heuristic $h(\cdot)$ requires estimating the relationship between the data characteristic and the future quality of training. The inputs for $h(\cdot)$ are categorized into a high-level meta-information as $\boldsymbol{\Pi}_{\text{meta}}$, and an intrinsic representation, denoted as $\boldsymbol{\Pi}_{\text{intrinsic}}$. In this paper, we examine the components of evaluating-exploring-evolving loop based on the formalization provided. Specifically, we focus on evaluating the data to derive $\boldsymbol{\Pi}_{\text{intrinsic}}$ and $\boldsymbol{\Pi}_{\text{meta}}$; exploring for the rule $\boldsymbol{\pi}^\star$ to estimate the high-quality distribution based on the model state $s$; and systematically generating and evolving more targeted data from $\mathcal{D}_{\text{unlabeled}}$.

### 2.1 COMPREHENSIVE INTERPRETABLE DATA DESCRIPTION

Some in-context (Yang et al., 2023; 2024b) or related data pruning works (Cao et al., 2023; Lu et al., 2024b) involve intrinsic descriptions, such as the visual attributes, resolution, clarity, color distribution, or text-related metrics, such as instruction/answer length, perplexity, and vocabulary. With the advanced zero-shot capabilities of LLMs and their multimodal extensions, many studies (Epasto et al., 2017; Lian et al., 2023; Chen et al., 2024c) propose using additional models to

replace manual evaluation. For example, Chen et al. (2024b) introduce ChatGPT, which improved performance efficiency compared to the Alpaca instruction tuning set, while Li et al. (2024a;b) analyze the instruction-following difficulty metrics. Due to storage constraints, some embedding-related methods are challenging to deploy in MLLM training. In this paper, we examine intrinsic and meta levels, presenting a multi-dimensional model zoo to comprehensively describe the dataset's nature, properties, and distribution.

## 2.2 DYNAMIC HIGH-QUALITY DATA SELECTION RULES

The data description serves as an input for the future quality estimation function $h(\cdot)$. At this point, designing high-quality data evaluation rule $\pi^\star$ based on the model state $s$ becomes crucial. Previous methods (Brown et al., 2020; Rae et al., 2021) are constrained by narrow data descriptions, resulting in unstable strategies influenced by random variables. They rely on costly empirical rules or deterministic mechanisms with manually set hyper-thresholds. For instance, some works (Raffel et al., 2020; Laurençon et al., 2022) consider word counts, statistical ratios, and multiple weight-based traversals. In recent years, increased interest has been in optimizing high-quality data rules based on model states and training environments (Xie et al., 2023c; Mingjian et al., 2022). Similar to the "no free lunch" theorem, no single rule is optimal for all situations. Approaches like DoReMi (Xie et al., 2023a) and DoGE (Fan et al., 2024) focus on optimizing the alignment of transferred distributions with the surrogate models. Some works (Biderman et al., 2023; Swayamdipta et al., 2020) start to consider searching for dynamic, high-quality data rules. In this paper, exploring is a critical step in ZOOPROBE for determining the form of the heuristic function $h(\cdot)$.

## 2.3 EFFICIENT TARGETED TRAINING DATA EVOLUTION

During the dataset expansion process, the candidate dataset can be sourced from collections or generated by additional MLLMs. Recently, the multimodal community has seen significant advancements in data engineering, starting with large image-text pair datasets such as Conceptual Captions (Sharma et al., 2018; Changpinyo et al., 2021), LAION (Schuhmann et al., 2021), and SBU (Ordonez et al., 2011). One type of approaches (Dai et al., 2023; Wang et al., 2023; Zhang et al., 2023a; Li et al., 2023b; Moon et al., 2023) tailor annotation information to create instruction candidates based on multimodal QA datasets, using strategies like template transformation, filtering, and deduplication to refine the data. Other methods (Li et al., 2023a; Yin et al., 2023; Wu et al., 2024; Zhao et al., 2023b; Zhang et al., 2023b; Wei et al., 2023) leverage the powerful zero-shot capabilities of LLMs and their extensions. For example, methods like LLaVA (Liu et al., 2023; 2024) convert annotated content into diverse task prompts to prompt ChatGPT, GPT-4, or GPT-4V for constructing visual instruction supervisions. However, such generative approaches, represented by methods like ALLaVA (Chen et al., 2024a), may lack targeted control over distributions, potentially harming diversity (Zhao et al., 2023a) and introducing bias based on the preferences of the generative MLLM (Xu et al., 2024). In contrast, LLMs such as Nemotron-4 (Adler et al., 2024) and SmolLM (Allal et al., 2024) pre-define detailed task guidelines, addressing these concerns more effectively. In this paper, ZOOPROBE investigates the relationship between more comprehensive data evaluation metrics and the high-quality rules to evolve accurately targeted new data.

## 3 METHOD: EVALUATING, EXPLORING, AND EVOLVING LOOP

In ZOOPROBE, we implement a dataset expansion framework using a heuristic search to enhance the training set for the next MLLM version. This process creates a dataset expansion tree, where each step generates multiple child-expanded dataset batches as leaf nodes from the existing training set. We *evaluate* and derive batch data descriptions for each child dataset, and *explore* selection rules for high-quality data based on these descriptions. Concurrently, we utilize these high-quality rules to systematically generate and *evolve* additional instructions and answers. ZOOPROBE with E3 loop facilitates the growth of child datasets and promotes new data generation.

### 3.1 EVALUATING DATA THROUGH EXTRA SMALL-SCALE MODEL ZOO

To comprehensively evaluate the data, we create a smaller-scale diversity model zoo that includes various models from different perspectives, such as topic classification (Li et al., 2023c), object

detection (Kirillov et al., 2023), depth estimation (Yang et al., 2024a), visual semantic representation (Dosovitskiy et al., 2021; Radford et al., 2021), and concept recall models (Jiang et al., 2024). We categorize description into *meta* properties and *intrinsic* attributes, where *meta* ones involve the interactions of high-level-evaluated information, and *intrinsic* ones refer to identifying visual objects, textual themes, and the detailed distribution of subjects. In ZOOPROBE, we conduct thousands of distribution assessments and evaluate over 50 dimensions to construct data descriptions.

**Extraction of meta descriptions.** Considering the efficiency of training set evaluation, ZOOPROBE adopts the textual and multimodal property ranker (Li et al., 2023c; Jiang et al., 2024) for a batch of data, denoted as $M_{\text{ranker}}^k$ for the critical dimensions, similar to the recommendation systems, where the ranker retrieves relevant meta-information for each data. Specifically, we extract prototypes $\mathcal{P}$ as the core representation that encodes the subjects and properties, then ZOOPROBE organizes and measures them according to their groups. At this stage, each batch of data retrieves a corresponding alignment degree from multiple meta prototypes. For instance, we offer keywords for over $5k$ topics, with a 5-level prototype for each property to annotate concepts from poor to excellent. Different rankers compute the alignment between the data and the meta prototypes. We formalize the meta-description as:

$$\mathbf{\Pi}_{\text{meta}}\left(\mathbf{d};\,\mathcal{P}\right) = \text{concat}\left\{\mathbb{E}_{\mathbf{x}\in\mathbf{d}}\left[M_{\text{ranker}}^k(\mathbf{x},\,\mathbf{p})_{\mathbf{p}\in\mathcal{P}}\right]\right\}_{k=1}^m, \qquad (2)$$

where $\mathbf{\Pi}$ represents the concatenated distribution of dimensional evaluations for various prototypes, as denoted by $\text{concat}\left\{\cdot\right\}$. The expectation vector $\mathbb{E}$ is computed based on the current expanded data batch $\mathbf{d}$ for each prototype $\mathbf{p}\in\mathcal{P}$.

**Extraction of intrinsic attribute.** Considering that the textual instructions in the training data are often generated based on visual content, we extract dominant fine-grained visual features. In detail, ZOOPROBE introduces specialized computer visual models (Dosovitskiy et al., 2021; Radford et al., 2021; Kirillov et al., 2023; Yang et al., 2024a) for object recognition, target detection, localization, depth estimation, and so on. Then it gathers expectation distribution on batch data, such as the count, location, area, depth, and other recognition information of objects in the foreground and background. Similarly, ZOOPROBE also collects some intrinsic attributes, such as the resolution and color style of the image, as well as the language and length of the text. Together, these distributions $\mathbf{\Pi}_{\text{intrinsic}}$, along with the meta-dimensions $\mathbf{\Pi}_{\text{meta}}$ mentioned above, form the description vector $\mathbf{\Pi} = \text{concat}\left\{\mathbf{\Pi}_{\text{meta}},\,\mathbf{\Pi}_{\text{intrinsic}}\right\}$ of the data batch $\mathbf{d}$. We have:

$$\mathbf{\Pi}_{\text{intrinsic}}\left(\mathbf{d}\right) = \text{concat}\left\{\mathbb{E}_{\mathbf{x}\in\mathbf{d}}\left[M_{\text{CV}}^{k'}(\mathbf{x})\right],\ldots,\mathbb{E}_{\mathbf{x}\in\mathbf{d}}\left[\text{Attr}^{k''}(\mathbf{x})\right],\ldots\right\}, \qquad (3)$$

where $M_{\text{CV}}$ represents a fine-grained visual model, and $\text{Attr}$ represents the intrinsic attributes of the data itself. In ZOOPROBE, during each step of selecting the next batch in the expansion tree, the evaluation function measures the current nodes' quality based on the data description distribution. Experiments show that ZOOPROBE allows for dynamic adjustment of the missing dimensions in $\mathbf{\Pi}$ based on available resources, enabling partial evaluation within the model zoo.

For evaluation dimension taxonomy, some in data description $\mathbf{\Pi}$ belong to *allocation*, manifesting as the distribution of themes or objects. Others include information about *properties*, expressed as high-level meta-descriptions, such as diversity, dependency, guidance, stability, ambiguity, and other cognitive interactions. On the other hand, high-quality data maintains specific ratios across some *scattered* dimensions. For example, the optimal balance between text-only and image-text instructions varies by context. Additionally, the language styles differ based on the situation. Meanwhile, low-quality data is detected and pruned in some other *filtered* dimensions, such as in dimensions related to data annotation quality, social biases, violence, and pornography.

## 3.2 EXPLORING HIGH-QUALITY SELECTION RULES BASED ON MODEL STATE

In section 2, we formalize the setting for high-quality data expansion and frame it as a search process on the expansion tree. Equation 1 describes the quality function which estimates the candidates of the next step. This function consists of the reward-so-far and heuristic estimate of future training quality, denoted as $r(\,\cdot\,;\,M)$ and $h(\,\cdot\,;\,\pi^\star)$, where $\pi^\star$ is a high-quality rule based on model state $s$. This subsection begins by defining the model state $s$ within the current training set $\mathcal{D}^{\text{cur}}$, followed by an explanation of the reward-so-far function. We then elaborate on the concept of high quality, which introduces the construction of the heuristic estimate $h(\,\cdot\,\pi^\star)$. Next, we detail how

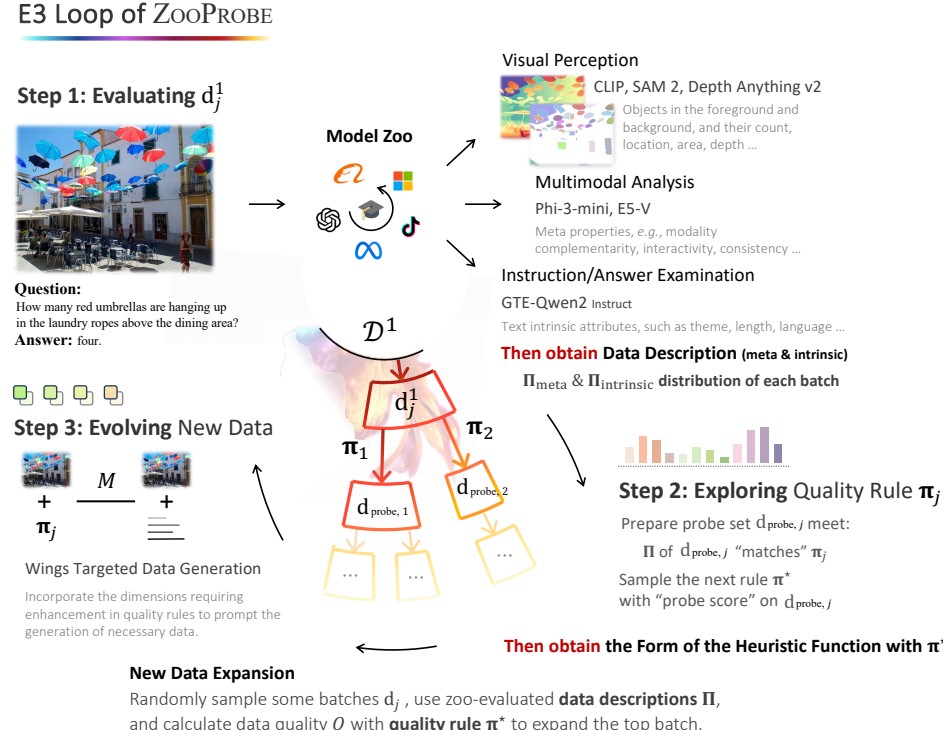

Figure 2: Visualization of Evaluating-Exploring-Evolving (E3) Loop.

ZOOPROBE leverages data descriptions $\mathbf{\Pi}$ to identify $\pi^\star$. Finally, we integrate this with the reward-so-far element and summarize the data qualities captured by the Equation 1.

**Reward-so-far of data reflected in model state.** We consider the performance of the current trained MLLM and the ratio of the learning speed as metrics to assess the reward so far. Formally:

$$s = r\left(\mathbf{d};\, M\right) = \text{Perf}\left(M_{\text{fine-tuned on } \mathbf{d}}\right) + \alpha \frac{\text{Step}_{\text{convergence}}}{\text{Step}_{\text{training}}}\, . \tag{4}$$

We sample a subset from the classic benchmarks to evaluate performance and use the percentage of convergence steps to represent the learning speed, where $\alpha$ is the coordination factor.

**High-quality conditional on the model state with the existing training data.** Considering that the evaluation results in Equation 2 and Equation 3 are expressed in the form of a distribution of expectation, ZOOPROBE naturally tends to describe high-quality selection rule $\pi^\star$ as a similar interpretable distribution. For the newly added batch $\mathbf{d}$, apart from some dimensions that are *filtered* as previously mentioned in subsection 3.1, most *scattered* dimensions exhibit a favorable distribution. We clarify that these *scattered* dimensions do not maintain a fixed ratio during dataset expansion. At each step, they are influenced by the current data $\mathcal{D}^{\text{cur}} = \mathcal{D}^1 + \mathbf{d}^1 + \mathbf{d}^2 + \mathbf{d}^3 + \cdots$. Some efforts (Tong et al., 2024) aim to avoid long-tailed, biased distributions, attempting to disperse the expectation on dimensions to achieve a sufficiently diverse dataset. We denote a batch $\mathbf{d}_{\text{diverse}}$ that makes the current dataset $\mathcal{D}^{\text{cur}}$ more diverse. Using Kullback-Leibler divergence to measure the difference, we have:

$$h\left(\mathbf{d}_{\text{diverse}};\, \mathcal{U}, \mathcal{D}^{\text{cur}}\right) = D_{\text{KL}}\left(\mathbf{\Pi}\left(\mathcal{D}^{\text{cur}}\right) \| \mathcal{U}\right) - D_{\text{KL}}\left(\mathbf{\Pi}\left(\mathcal{D}^{\text{cur}} + \mathbf{d}_{\text{diverse}}\right) \| \mathcal{U}\right) \geqslant 0\, . \tag{5}$$

Achieving greater diversity, namely the distribution of batch $\mathbf{d}$, is the sampling trend for high-quality data. However, the optimal distribution may not always be uniform. For instance, certain challenging concepts, such as "person", which interact intricately with their context in multimodal environments, should have more coverage in the training set. The primary objective of the rule exploration is to determine the optimal distribution $\pi^\star$ of the upcoming batch through the model state $s$, as it reflects the reward and state of current set $\mathcal{D}^{\text{cur}}$.

**Exploration of heuristic estimate (high-quality rule).** Firstly, we conduct $r$ random perturbations based on $\mathcal{U}$ to obtain $\boldsymbol{\pi}_1, \boldsymbol{\pi}_2, \cdots, \boldsymbol{\pi}_r$. Simultaneously, we sample corresponding batches, which we refer to as the probe set, *i.e.*, $\mathbf{d}_{\text{probe}, 1}, \mathbf{d}_{\text{probe}, 2}, \cdots, \mathbf{d}_{\text{probe}, r}$, subject to $h\left(\mathbf{d}_{\text{probe}, j}; \boldsymbol{\pi}_j, \mathcal{D}^{\text{cur}}\right) \geqslant 0$. Next, for the exploration step, the model is trained on the union of all probe sets $d_{\text{probe}, j}$ and the current dataset $\mathcal{D}^{\text{cur}}$ to explore the reward. The adopted optimal distribution $\boldsymbol{\pi}^\star$ prioritizes sampling based on $\tau$-based $\mathrm{softmax}$ of so-far-rewards. The sampling weight $w$ for $\boldsymbol{\pi}_j$ is:

$$w_{\boldsymbol{\pi}_j} = \mathrm{softmax}^\tau \left(r\left(\mathbf{d}_{\text{probe}, j}; M\right)\right), \text{ s.t. } h\left(\mathbf{d}_{\text{probe}, j}; \boldsymbol{\pi}_j, \mathcal{D}^{\text{cur}}\right) \geqslant 0. \quad (6)$$

The distribution $\boldsymbol{\pi}_j$ associated with the higher so-far-reward of $\mathbf{d}_{\text{probe}, j}$ suggests an increased likelihood of exploitation. The weighted sampling also maintains the exploration of other distributions.

We provide cost trade-off solutions for probe-based fine-tuning by: **1**) fine-tuning with parameter-efficient methods, and **2**) employing the method for pre-assess the performance of the model after fine-tuning on the probe set. We follow Nguyen et al. (2020), introducing log expected empirical prediction as a proxy metric for efficiently estimating the transferability between the base MLLM and the probe set. It only requires inferring the target set $\{\mathcal{D}^{\text{cur}} + \mathbf{d}_{\text{probe}, j}\}$ once through the MLLM.

Next, the data tree is expanded within the A$^\star$ framework. After sampling $\boldsymbol{\pi}^\star$ from Equation 6, we refine the form of the heuristic function in Equation 5. Meanwhile, the reward-so-far function, as given in Equation 4, is expressed in terms of the parent nodes. Subsequently, ZOOPROBE randomly samples some expansion batches and calculates the quality with the Equation 1.

### 3.3 EVOLVING NEW DATA FROM HIGH-QUALITY GUIDELINES

In the data expansion tree, ZOOPROBE can sample the child datasets from existing labeled collections of visual instructions. However, the wild scale can be thousands of times larger than the sample size. Like related works, ZOOPROBE enhances sampling efficiency and introduces additional pre-trained MLLMs to generate new data. Furthermore, ZOOPROBE creates more high-quality data by considering the high-quality rules identified in Equation 6, targeting the dimension gaps in the existing training set. It first identifies missing features in the current distribution and uses a representative set with rule-based prompts to generate detailed macro task information. This output then serves as a subsequent prompt for generating new instructions and responses.

**Macro task prompt as detailed data generation guidelines based on high-quality rules.** As illustrated in Figure 2, ZOOPROBE begins by integrating an additional MLLM to outline the high-quality components missing in the current training distribution. For instance, if specific themes, such as mathematics or medicine, lack sufficient instructions or properties like inadequate spatial reasoning or interaction with visual information, the additional MLLM generates relevant sub-themes and detailed descriptions.

**New data generation with macro task prompts and representative examples.** ZOOPROBE retrieves images based on specified rules and generates new instructions and responses using macro task prompts and representative examples. This data is then added to an iteration of the expansion tree and evaluated in the E3 loop. If it meets quality standards, the training set is enhanced. ZOOPROBE maintains a dynamic balance between sampling from wild collections and generating new data, ensuring a rich and diverse training data evolution.

### 3.4 LEARNING A DATA QUALITY SCORER ON EXPANSION TREE

In the E3 loop of data expansion, many filtered and discarded training sets are marked. We deploy an MLLM ranker to learn a quality score for the training data. Specifically, we construct both positive and negative samples, adding the prompt: "Please consider the ... and answer whether it is suitable for training with just 'yes' or 'no'." We use the cross-entropy loss to ensure the probability of the model outputting 'Yes' is high for positive samples. In subsequent heuristic expansions, the MLLM scorer can replace the evaluating and exploring processes, directly assessing the data quality.

## 4 EXPERIMENTS

In this section, we first outline the setting, architecture, and details of the E3 loop. Following that, we analyze how ZOOPROBE improves performance and efficiency, along with the effectiveness of

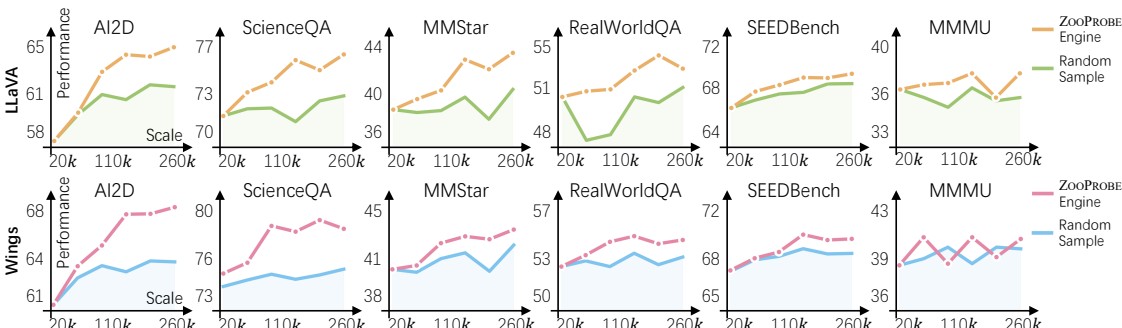

Figure 3: **Comparison of scaling-laws performance between the ZooPROBE data engine and random sampling across major benchmarks.** The horizontal axis indicates the scale of training data in increments of $20k$ to $260k$, *i.e.*, from $20k$, $60k$, $110k$, $160k$, and $210k$, to $260k$. The vertical axes are calibrated to maintain a consistent range of 8, facilitating relative comparison. The first row and second row are the results for LLaVA (Liu et al., 2023) and Wings Zhang et al. (2024), respectively, where the yellow and pink lines indicate training with ZooPROBE.

the MLLM-data-scorer in assessing data quality as in subsection 3.4. In ablation studies, we evaluate the acceptable levels of zoo evaluate time, incomplete descriptions, or the absence of rule search.

**Details of the dataset expansion setting**: To simulate the original dataset expansion, we randomly sample $10k$ from the LLaVA-NeXT (Liu et al., 2024) fine-tuning instructions as the base dataset. Each expansion batch consists of $1k$ samples. After 50 heuristic updates, we add $50k$ samples and updated the existing training set. Subsequently, we perform quality rule sampling and updates based on Equation 6 to refine the heuristic function for the next 50 iterations.

**Details of MLLM architecture, training, and benchmark Evaluation**: In our experiments, we utilize the LLaVA (Liu et al., 2023) architecture and its extended version, Wings (Zhang et al., 2024). LLaVA employs a visual encoder with a connector to extract image features, which are then combined with instruction features for input into the LLM. Compared to LLaVA, Wings adds an attention compensation module. We utilize Llama3.1-Instruct (Dubey et al., 2024) for all LLM parts and SigLIP (Zhai et al., 2023) as the visual encoder. Starting from pretrained MLLM of LLaVA-1.5 (Liu et al., 2023), ShareGPT4V (Chen et al., 2023), and ALLaVA-Caption (Chen et al., 2024a), we fix the visual encoder and fine-tuning the LLM and connector with a learning rate of $2e^{-6}$ and $1e^{-5}$, respectively. All training is on $8\times$ A100 GPUs. We evaluate on MMMU (Yue et al., 2023), MMStar (Chen et al., 2024c), RealWorldQA (x.ai, 2024), AI2D (Kembhavi et al., 2016) for test, ScienceQA (Lu et al., 2022) for test, and SEEDBench (Li et al., 2023d).

**Details of *evaluating*: model zoo and dimensions**: We design both the *meta* and *intrinsic* data characteristic dimensions, introducing multi-faceted textual, visual, and multimodal models to evaluate the data comprehensively. As shown in Figure 2, GTE-Qwen2$_{\text{Instruct}}$ (Li et al., 2023c) performs evaluations based on instructions and answers, covering dimensions such as linguistic complexity, potential ambiguity, conceptual depth, knowledge coverage, logical structure, inferred established information, emotional subjectivity, temporal stability, and practical applicability. E5-V (Jiang et al., 2024) encompasses multimodal characteristics including annotation quality, image quality, the difficulty of object recognition in images, text comprehension levels, visual content diversity, scene complexity, dependency on visual and textual content, multimodal interactivity, requirements for external knowledge, cognitive load, context interpretability, detail orientation, novelty and unseen concepts, as well as specific time series, action dynamics, emotion recognition, mathematical concept coverage, interpretation of postures and gestures, spatial reasoning, and interpretability of graphic elements. Additionally, CLIP (Radford et al., 2021) and SAM (Kirillov et al., 2023) work together to complete visual object recognition, including counting, location, and size identification, while Depth-Anything-V2 (Yang et al., 2024a) estimates the depth of objects. We also extract the language and token sequence length of the instructions and answers, as well as the images' brightness, color palette, resolution, and clarity.

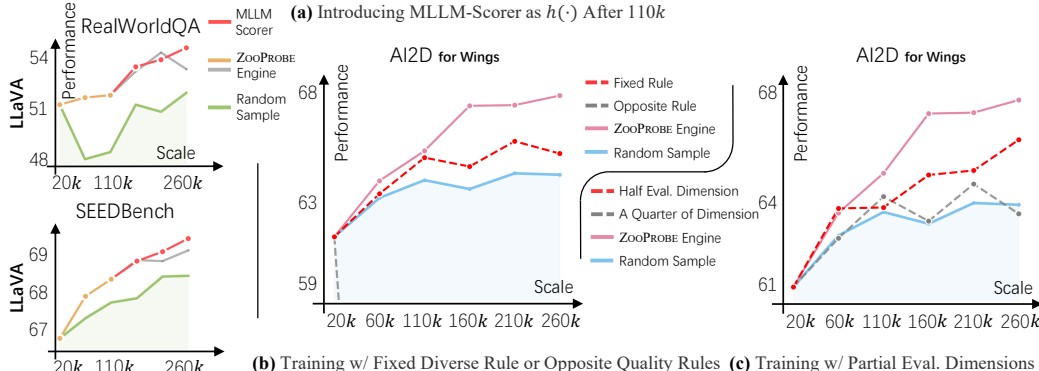

Figure 4: **Ablation studies of the heuristic function determined by the quality rule.** (a) We employ an additional learned data quality, MLLM-Scorer, as the assess function for training LLaVA, introduced after the 110k scale and is highlighted in red. (b) We examine a fixed diverse strategy (a red dashed line) with a strategy that is opposite to ZOOPROBE's (a gray dashed line). (c) We analyze the effects of partial dimensional in the data description, considering half-dimensional and quarter-dimensional setups marked with red and gray dashed lines, respectively.

| Step Training Time *v.s.* Evaluating $\Pi_{meta}, \Pi_{intrinsic}$ | Time Cost |
|---|---|
| **On Expansion Dataset:** | |
| Step Training | ~72x |
| **Evaluating On Expansion Dataset:** | |
| Fine-grained $\Pi_{meta}$ | ~6x |
| Retrieving $\Pi_{meta}$ | ~1x |
| Extracting $\Pi_{intrinsic}$ | ~0.1x |

**(a)** Intuitive Comparison of Training *v.s.* Evaluating

| Step Training Time *v.s.* Exploring $\pi_1, \pi_2, \cdots, \pi_r$ | Time Cost | GPU Memory Overhead |
|---|---|---|
| **On Expansion Dataset:** | | |
| Step Training | ~450x | ~40x |
| **On Probe Set:** | | |
| Fine-tuning (Full Parameter) | ~15x | ~40x |
| Fine-tuning (LoRA) | ~2.5x | ~10x |
| LEEP Transferability | 1x | 1x |

**(b)** Intuitive Comparison of Training *v.s.* Exploring Rules

Figure 5: **Comparison between the running overhead of ZOOPROBE and model training per step.** (a) We compare the costs across three types of dimensions during the evaluation step. The average evaluation cost per iteration is approximately 1/50 of the step training cost. (b) We compare the overhead of rule exploring on the probe set every 50 steps of dataset expansion. The average exploring overhead is about 1/20 of the training overhead. We extend and probe five branches.

**Details of *exploring*: rule sampling with reward-so-far evaluation.** In Equation 6, we investigate and explore rules after conducting 50 iterations of A$^\star$ search. During this process, we refine the heuristic by perturbing the uniform distribution $\mathcal{U}$. This perturbation, utilizing the $D_{KL}$-based heuristic, disrupts certain dimensions that are normally required to maintain direction towards $\mathcal{U}$ during optimization. To evaluate the effectiveness of different rules, we sample a probe set of $10k$ and calculate the cumulative reward. As shown in subsection 3.2, for datasets with fewer than $200k$ parameters, we perform full parameter fine-tuning, and for larger datasets, we adopt parameter-efficient LoRA-tuning (Hu et al., 2022).

**Details of *evoluting*: data sampling collections, data generation prompts, and conditions**: As discussed in subsection 3.2, most of the expanded data is sampled from existing data collections. We consider a massive fine-tuning pool, including ShareGPT4V (Chen et al., 2023), LLaVA-1.5-finetune (Liu et al., 2023), LLaVA-NeXT (Liu et al., 2024), ALLaVA-finetune (Chen et al., 2024a), and Cambrian-10M (Tong et al., 2024), which contains subclasses such as language, OCR, counting, code, math, and science, all under $300k$ threshold. We filter the general, language, and OCR subclasses and obtain about $3.5m$ samples. When a sampling batch exceeds 1,000 but fails to meet the quality standards, we initiate a data generation process. This involves generating macro task prompts to address the identified gaps, facilitating more accurate new instruction and response generation.

**Comparative analysis of different architectures and scaling.** As detailed in section 4, the first and second rows contrast the scaling performance of LLaVA and Wings against the benchmarks

set by random data sampling and the expanded dataset used by ZOOPROBE. Overall, ZOOPROBE significantly outperforms the random sampling strategy across all benchmarks for both MLLM architectures. One key advantage of ZOOPROBE is its ability to tailor exploration rules based on model state and training situations. For instance, between the $60k$ to $260k$ scale range, LLaVA and Wings apply different heuristics, resulting in varied data expansion strategies. Our experiments reveal that a dynamic quality rule derived from the probe set-based search, defined in Equation 6, consistently delivers superior performance across different architectures and model states. As we progress to larger scales, particularly beyond the $110k$ mark, ZOOPROBE exhibits continuous performance enhancements, further amplifying its advantage over random sampling. This trend indicates that strategically exploring rules based on existing data not only accumulates training quality but also boosts performance as dataset sizes increase.

**Analysis of Detailed Benchmarks.** When comparing across several benchmarks, including AI2D, ScienceQA, MMStar, and RealWorldQA, ZOOPROBE shows marked improvements across various domains. Remarkably, despite having only about one-third of LLaVA-1.5's training volume ($260k$), its performance on the AI2D datasets is nearly equivalent. This indicates that our data engine enhances MLLMs' capabilities and serves as an automated pipeline for high-quality data adaptation.

**Training of an additional MLLM-Scorer to serve as the heuristic estimation.** In Figure 4 (**a**), we present the results achieved after integrating MLLM-Scorer with over $110k$ data points. Compared to ZooProbe, MLLM-Scorer performs at a similar level, highlighting its effectiveness in evaluating data quality and showcasing its robust performance.

**Ablation studies on the heuristic functions, evaluated dimensions, and efficiency** of ZOOPROBE. In Figure 4, we validate the effectiveness of ZOOPROBE's dynamic quality rule, demonstrating its ability to identify data that truly benefits model training. Next, we focus on the efficiency of ZOOPROBE, controlling the number of dimensions during evaluation to see if we can maintain performance while only considering the partial dimensions. In Table 5, we provide a detailed comparison of the overhead of ZOOPROBE relative to the training process.

- Why is the dynamic exploring rule in ZOOPROBE effective? In Figure 4 (**b**), we first establish that the opposite rule leads to performance degradation. Although fixed diverse rules emphasize data variety and outperform random sampling, they are still weaker than ZOOPROBE. This suggests that the data requirements for the model change dynamically; certain less utilized dimensions of data may become less important as the iteration progresses.

- In resource-limited situations, can ZOOPROBE maintain performance by using only a subset of evaluation dimensions effectively? As shown in Figure 4 (**c**), we randomly sample half or a quarter of the evaluation dimensions and find that high-level evaluation information provided crucial insights for exploring high-quality rules and optimizing performance. While using partial dimensions results in some performance decline, it still outperforms random sampling.

- We compare the costs of ZOOPROBE's evaluation and exploration part to step training. In Table 5 (**a**), the fine-grained data description refers to those with derivational relationships, such as the high-level meta relationship concerning "quality". This part requires more time for inference outputs but constitutes only about one-sixth of all the dimensions. Additionally, in Table 5 (**b**), we present the costs of three different methods for estimating the quality of the probe set.

## 5 DISCUSSION AND CONCLUSION

In this paper, ZOOPROBE continuously generates high-quality data through the proposed evaluating-exploring-evolving loop, beating the performance barrier of scaling laws under equivalent training costs. The ZOOPROBE's model zoo comprehensively evaluates data characteristics and, combined with heuristic exploration strategies, explores dynamic quality rules based on varying model states on probe set. Furthermore, ZOOPROBE can specifically generate high-quality data, address distribution gaps, and create an adaptive, universal data engineering framework. Data deconstruction in ZOOPROBE is similar to combinatorial dimensions in zero-shot learning, but ZOOPROBE explores properties from a higher-level perspective. Future directions may include efficiently establishing the relationship between latent space and training quality while introducing learnable factors to accelerate the search process. We will also investigate the safety of data generation, ensuring quality while avoiding bias and considering transferability across different model scales.

## ACKNOWLEDGMENTS

This work is partially supported by NSFC (62376118, 62250069), Key Program of Jiangsu Science Foundation (BK20243012), Collaborative Innovation Center of Novel Software Technology and Industrialization.

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

## A   APPENDIX

We sample some of the top-performing data on some evaluation dimensions, as shown below:

## Action Dynamics

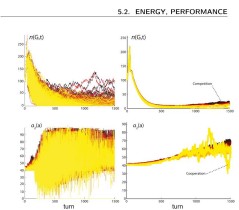

Question: <image>\nWhat are the different initializations?

Answer: The different initializations are related to mobility, competition and cooperation.

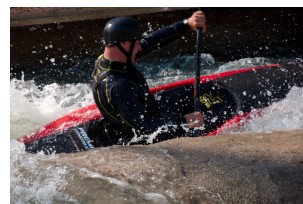

Question: <image>\nAnalyze the image in a comprehensive and detailed manner.

Answer: The image captures a thrilling moment of a kayaker navigating through turbulent waters. The kayaker, clad in a green wetsuit and a black helmet, is seen skillfully maneuvering a red and black kayak, which sports a yellow logo on its side. The water around the kayak is white and frothy, indicating the intensity of the rapids. The kayaker uses a paddle to steer through the tumultuous water, demonstrating a high level of skill and control. The image is taken from a side perspective, adding a sense of depth and dynamism to the scene. In the background, a large rock formation looms, providing a stark contrast to the churning waters in the foreground. The image does not provide any identifiable information about the landmark referred to as 'sa_108'.

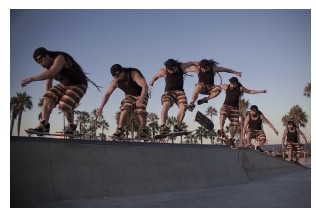

Question: Is the man performing tricks on a skateboard in the image?\n<image>

Answer: Yes, the man in the image is performing tricks on a skateboard.

Question: What type of image is used to capture the man's skateboard tricks?

Answer: A time-lapse or a series of multiple pictures has been used to capture the man's skateboard tricks, effectively showing the movements and progression of the trick he is performing.

Question: Is there only one image, or are there multiple images combined to show the skateboarder's movements?

Answer: There are multiple images combined to show the skateboarder's movements while he performs the trick.

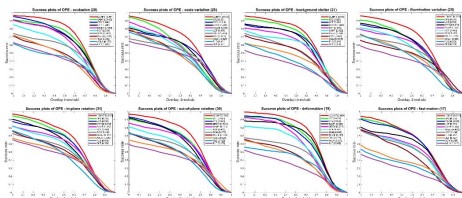

Question: <image>\nWhat are the key takeaways from the graph?

Answer: The key takeaways from the graph are that the proposed tracker outperforms the state-of-the-art trackers on almost all challenge attributes. In particular, the proposed tracker performs significantly better than all trackers on the occlusion attribute since it includes a re-detection module which can re-acquire the target in case the tracker fails even in cluttered environments by removing clutter using GM-PHD filter. Similarly, the proposed tracker also outperforms other trackers on the scale variation attribute since our tracker elegantly estimates the scale of the tracker at the newly estimated target positions.

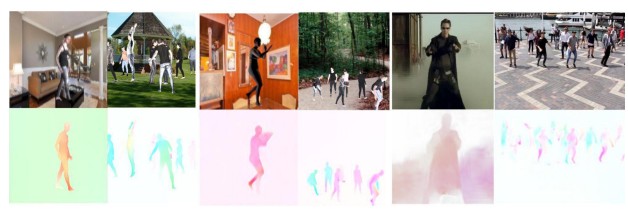

Question:   <image>\nWhat pattern is observed in the movements of the individuals across the images? A) The individuals are progressively moving faster. B) The individuals are standing still. C) The individuals are engaging in a coordinated activity. D) The individuals are randomly scattered without any clear pattern.

Answer: The images show a sequence of movements where the position and stance of individuals change systematically, suggesting a coordinated activity like a dance or exercise routine. Therefore, the correct answer is C) The individuals are engaging in a coordinated activity..

## Accuracy of OCR Data

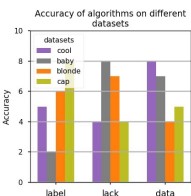

Question: <image>\nWhat is the accuracy of the algorithm data in the dataset cap?\nAnswer the question using a single word or phrase.

Answer: 5

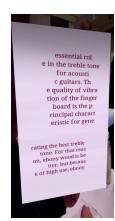

Question: <image>\nCan you accurately transcribe the text from the provided image for me?

Answer: essential rol e in the treble tone for acousti c guitars. Th e quality of vibra tion of the finger board is the p rincipal charact eristic for gene rating the best treble tone. For that reas on, ebony wood is be tter, but becaus e of high use, ebony

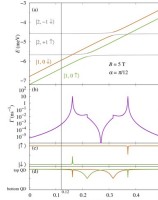

Question: <image>\nWhat magnetic field strength is depicted in the energy diagram of the figure? A) 1 Tesla B) 3 Tesla C) 5 Tesla D) 10 Tesla

Answer: The figure has a label \"B = 5 T\" which indicates the magnetic field strength used in the energy diagram is 5 Tesla. Therefore, the correct answer is C) 5 Tesla.

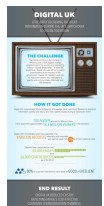

Question: <image>\nHow many star symbols are there near the text \"80%\"?\nAnswer directly.

Answer: 5

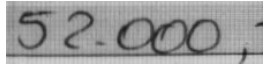

Question: <image>\nWhat are the number in the image?

Answer: 52000

## Complexity of Visual Scenes

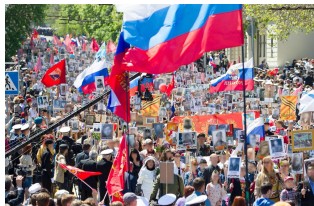

Question: <image>\nAnalyze the image in a comprehensive and detailed manner.

Answer: The image captures a vibrant scene of a large parade, teeming with people. The crowd, a sea of individuals, is marching with a sense of unity and purpose. They are holding up various flags and banners, the Russian flag and the flag of the Soviet Union being particularly prominent, adding a splash of color to the scene. The people are also holding up photographs of soldiers and other historical figures, perhaps in remembrance or celebration. The perspective of the photo is from a high angle, providing a bird's eye view of the crowd, the flags they carry, and the photographs they hold up. This perspective allows us to appreciate the scale of the event and the number of participants. The background of the image is a mix of trees and buildings, providing a contrast to the crowd and highlighting the urban setting of the parade. The landmark identifier \"sa_15895\" does not provide additional information about the specific location or event.

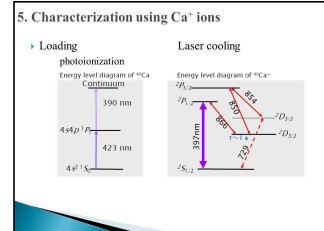

Question: <image>\nBased on the molecular structures and data charts displayed in the background, what type of research or experiment might the scientist be conducting?

Answer: The presence of molecular structures and detailed data charts on the digital display suggests that the scientist is likely involved in molecular biology or biochemistry research. The use of a pipette and petri dish indicates he might be conducting experiments related to cell culture, genetic analysis, or drug development. The complexity of the displayed data implies advanced research, possibly in areas like gene editing, protein synthesis, or pharmaceutical testing.

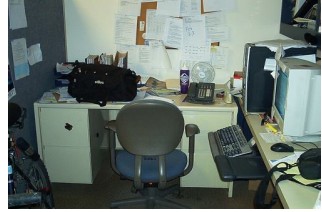

Question: <image>\nAssess the quality of the image by describing its visual complexity.

Answer: The image has high visual complexity, with multiple objects, overlapping boundaries, and various sizes and orientations of the items.

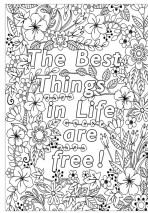

Question: <image>\nConsidering the variety of floral patterns presented, which specific flower or leaf design appears the most frequently throughout the composition, and what could be the significance of this repetition within the context of a coloring page?

Answer: The five-petal blossoms seem to be the most frequently recurring floral design in the composition. The repetition of this specific flower could serve several purposes: it provides a consistent element that ties the diverse patterns together, creating visual harmony; it offers a simpler shape for coloring, which could be appealing for those who want repetitive and meditative coloring tasks; and aesthetically, it could represent a theme of unity or abundance, which is often associated with nature and floral motifs in art.

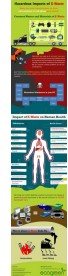

Question: <image>\nhow many common materials of e-waste are given in the chart?\nRespond concisely.

Answer: 12

Question: how many substances from the chart affect liver?

Answer: 5

Question: what are the substances inside PC that causes liver cancer?

Answer: arsenic, plastics/PVC

## Contextual Interpretation

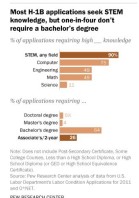

Question: <image>\nWhat are some skills required to H-1B applications?

Answer: Most H-1B applications ( 75 % ) require high - level computer knowledge , and roughly half require significant engineering and math skills . High - level scientific knowledge is less commonly requested (11%)

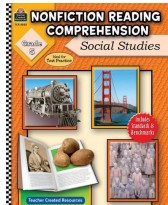

Question: <image>\nWhat is the title or main subject of the book mentioned in the image?

Answer: The book focuses on Nonfiction Reading Comprehension in social studies, ideal for test practice. It targets Grade 3 students and includes standards benchmarks. The book is created by Teacher Created Resources.

Question: Does the image indicate the content of the book?

Answer: Yes, the image suggests that the book contains nonfiction reading comprehension materials relating to social studies topics. This could potentially include information on cities, bridges, and other aspects of human society and infrastructure.

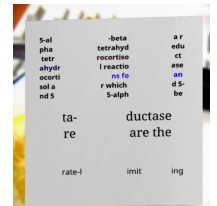

Question: <image>\nCan you read and provide the text displayed in the image?This photo seems to have some interesting text. Can you extract and type it out for me?

Answer: 5-al pha tetr ahydr ocorti sol a nd 5 -beta tetrahyd rocortiso l reactio ns fo r which 5-alph a r edu ct ase an d 5- be ta- re ductase are the rate-l imit ing

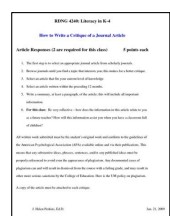

Question: <image>\nWhat is the main topic discussed in the text?

Answer: he main topic discussed in the text is RDNG 4240 Literacy in K-4 and guidelines on how to write a critique of a journal article for the class.

Question: How many points are each article response worth?

Answer: Each article response is worth 5 points.

Question: How many article responses are required for the class?

Answer: Two article responses are required for the class.

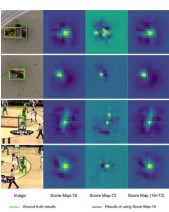

Question: <image>\nWhat does the \"Score Map\" in the figures represent in the context of the scientific analysis being performed? A. The likelihood of the presence of an object in various locations within the image. B. The color intensity variations across the image. C. The temperature distribution captured by a thermal camera. D. The altitude variations on a topographic map.

Answer: The Score Map images represent heatmaps where higher intensities indicate a higher likelihood of the presence of an object at that location. This is indicated by the concentration of intense colors at certain points corresponding to the object's location in the ground-truth images. Therefore, the correct answer is A.

## Dependence on Textual Information

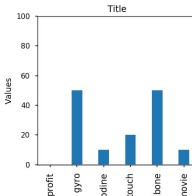

Question: <image>\nWhy do you think the title of the chart is simply 'Title'? What could this imply?

Answer: The generic 'Title' could suggest that this chart is a template or example and has not been customized for a specific presentation. It likely serves as a placeholder, indicating where a descriptive title relevant to the data should be added.

Question: How important is it to have a descriptive title for a chart?

Answer: Having a descriptive title is crucial because it provides context for the data and assists viewers in quickly understanding the chart's purpose. It can convey the subject or scope of the data, making the chart more effective and accessible for interpretation.

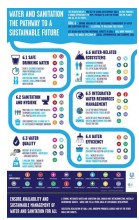

Question: <image>\nWhat percentage of global population depends on ground water for drinking?\nAnswer directly.

Answer: 50%

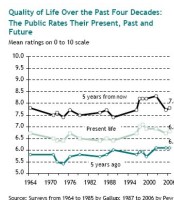

Question: <image>\nWhat is the perception of people about their lifestyle over the years and now?

Answer: in the 2006 Pew survey ( taken this summer ) , Americans on average gave a 6.1 score to their lives five years ago ; a 6.8 score to their present quality of life ; and a 7.8 score to the lives they expect to be leading five years from now . That adds up an aggregate average of 1.7 rating points worth of forward progress between five years ago and five years hence

Question: <image>\nWhat do variational methods allow the estimation of? Directly refer to the text in the image to answer this question.

Answer: Variational methods allow estimation of hierarchical discrete choice models in a small fraction of the time required for MCMC.

Question: What is the value proposition of variational methods? Directly refer to the text in the image to answer this question.

Answer: The value of the variational approach is greatest when sub-sampling of data is ill-advised.

Question: According to the authors, should MCMC be abandoned in favor of variational methods? Provide an answer to this question by directly using the text provided in the image.

Answer: We emphasize that we do not advocate abandoning MCMC in favor of variational methods.

## Detail Orientation

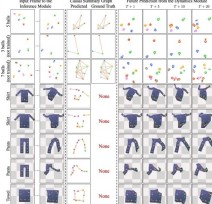

Question: <image>\nWhat patterns can be inferred about the interaction among the five balls from the input frame to the inference module and the causal summary graph? A. The balls are not interacting with each other at all. B. There is a one-directional interaction from one ball to all others. C. The interactions are complex and multi-directional among the balls. D. The balls interact only in a linear fashion.

Answer: The correct answer is C because the causal summary graph for the five balls shows multiple lines connecting each ball, indicating complex and multi-directional interactions. The lines represent interactions, and since they are not unidirectional or absent, options A, B, and D are incorrect. Therefore, the correct answer is C.

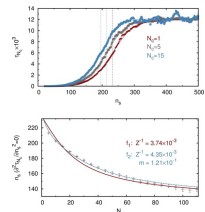

Question: What can be deduced about the precision of measurements for $N_c = 5$ in the first graph? A. High precision, as indicated by the tightly packed data points B. Moderate precision, as indicated by the spread of data points C. Low precision, as indicated by the widely spread data points D. Precision cannot be determined from the graph\n<image>\n

Answer: The first graph shows the data points for $N_c = 5$ closely following the trend line with little deviation, indicating that the measurements were consistently close to the expected values. This suggests a high level of precision in the measurements, making option A the correct answer. Therefore, the correct answer is A.

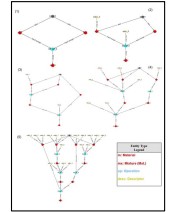

Question: <image>\nWhat are the different difficulty levels?

Answer: The difficulty levels are based on the number of actions in the quest. The easiest quests have only a few actions, while the hardest quests have many actions.

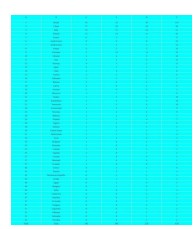

Question: <image>\nWhat is the total number of medals between south korea, north korea, sweden, and brazil?\nAnswer the question using a single word or phrase.

Answer: 150.

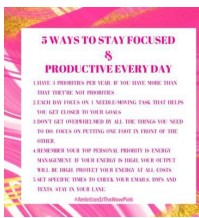

Question: <image>\nWhat are the 5 ways to stay focused and productive mentioned in the image?

Answer: The 5 ways to stay focused and productive mentioned in the image are:\n1. Have 3 priorities per year. If you have more than that, they're not priorities.\n2. Each day, focus on 1 needle-moving task that helps you get closer to your goals.\n3. Don't get overwhelmed by all the things you need to do. Focus on putting one foot in front of the other.\n4. Remember your top personal priority is energy management. If your energy is high, your output will be high. Protect your energy at all costs.\n5. Set specific times to check your emails, DMs, and texts. Stay in your own lane.

Question: How would you describe the aesthetic of the image?

Answer: The image has a pink and gold abstract background with a white frame, which gives off a modern and stylish appearance. The text is well laid out, easy to read, and complements the overall design while conveying the message effectively.

# Diversity of Visual Content

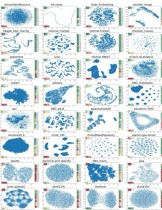

Question: <image>\nBased on the scatter plot for 'satellite image,' what could be inferred about the dataset? A) High dimensionality with clear separation B) Low variance among the data points C) Uniform distribution across different dimensions D) Data points are concentrated in a specific region -

Answer: The scatter plot for 'satellite image' shows data points spread out in a distinct, multi-directional pattern, indicating high dimensionality with some degree of separation between data points. Therefore, the correct answer is A.

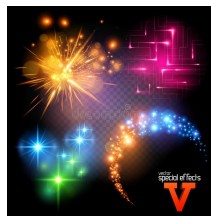

Question: <image>\nWhat might be the intended use or application of these vector special effects based on their design and composition?

Answer: Based on their design and composition, these vector special effects could be intended for use in digital media, such as graphic design, web design, or video production. They might be used to enhance visual appeal, simulate natural phenomena like fireworks or stars, or create dynamic backgrounds and overlays for various types of visual content. The bright colors and glowing effects would be particularly useful for projects that aim to grab attention or convey a sense of energy and vibrancy.

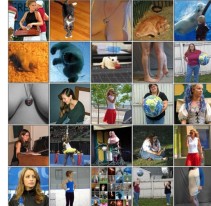

Question: <image>\nWhat is the main subject of the image?

Answer: The main subject of the image is a collage of photos.

Question: What types of photos are included in the collage?

Answer: The collage contains photos of persons, animals, and lockets.

Question: What type of question is being asked in the image?

Answer: There is no question present in the image; it is a collage of photos featuring persons, animals, and lockets.

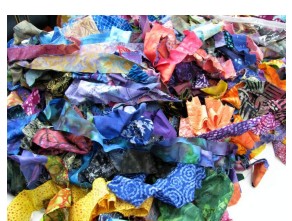

Question: <image>\nConsidering the assortment of fabrics, what could be inferred about the potential use or origin of these scraps?

Answer: The mix and disorder of the fabric scraps suggest they may be remnants from various sewing or quilting projects. The diverse patterns and colors could imply that they are offcuts from a place where multiple types of garments or textile products are made, such as a tailor's shop, a fashion design studio, or a crafts workshop. Their varied nature indicates they might be used for patchwork, crafting, or as fillers for art projects, given their vibrant and contrasting aesthetics.

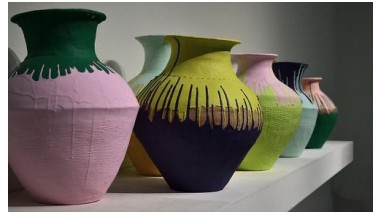

Question: <image>\nDescribe the diversity of colors and patterns on the vases in the image.

Answer: The vases have varied colors and patterns, such as green and pink, lime green and black, black and teal, terracotta and green, and yellow and purple, making each vase unique and attractive.

## Emotion Recognition

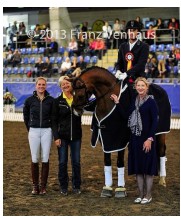

Question: <image>\nPlease describe the process of counting the number of people distinctly present in the provided image.

Answer: From the bounding box information, we can identify that there are at least five distinct individuals: a man sitting on the horse, a lady in a blue outfit, three women standing by the man on the horse, and people in the background.

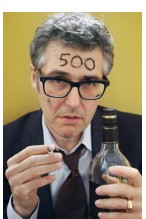

Question: <image>\nConsidering the objects the man is interacting with and the number on his forehead, what could be a possible scenario that explains the combination of these elements?

Answer: The scenario could be that the man has participated in a challenge or a bet, which is commonly marked by writing on one's skin with a marker. The number \"500\" might indicate a milestone, a score, or a target associated with the challenge. The presence of the cigarette and the bottle could suggest that this activity is taking place in a social setting where smoking and drinking are involved, perhaps a private gathering or a celebration. The man's serious expression and formal attire might indicate that this event is significant to him, or that he is in a moment of contemplation about the activity he is engaged in.

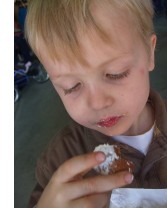

Question: are there any other people besides boy?\n<image>\nRespond concisely.

Answer: no

Question: how old does boy look?

Answer: 5

Question: is picture indoors?

Answer: hard to tell

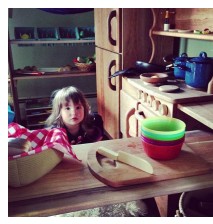

Question: <image>\nis child male or female?\nRespond concisely.

Answer: femal

Question: how old is girl?

Answer: 5

Question: what color is her hair?

Answer: brow

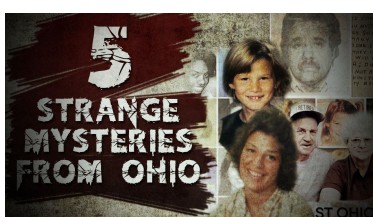

Question: <image>\nConsidering the theme of the image and the arrangement of the photographs, what might be the significance of the partially obscured photo in the top right corner, and how does it contribute to the overall narrative suggested by the image?

Answer: The partially obscured photo in the top right corner could represent an element of mystery or an unresolved aspect of a story. Its partial concealment behind the other photographs suggests that it may hold information that is not fully revealed or understood, contributing to the theme of mystery. In narratives, especially those dealing with unsolved mysteries or cases, incomplete information or partially revealed clues can be a deliberate device to engage the audience, prompting them to seek more information and become more involved in the unraveling of the story. Thus, the obscured photo adds to the enigmatic quality of the image and invites further inquiry into the mysteries presented.

## Mathematical Concept Coverage

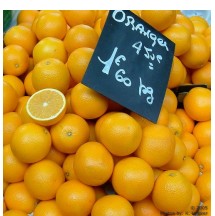

Question: <image>\nCritique the image's quality and provide suggestions for improvement.

Answer: The image quality is detailed and high, capturing various elements such as the oranges' textures, chalkboard sign with white lettering, and even a watermark. However, it could be improved by minimizing the glare of the light on the orange skin.

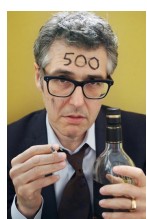

Question: <image>\nConsidering the objects the man is interacting with and the number on his forehead, what could be a possible scenario that explains the combination of these elements?

Answer: The scenario could be that the man has participated in a challenge or a bet, which is commonly marked by writing on one's skin with a marker. The number \"500\" might indicate a milestone, a score, or a target associated with the challenge. The presence of the cigarette and the bottle could suggest that this activity is taking place in a social setting where smoking and drinking are involved, perhaps a private gathering or a celebration. The man's serious expression and formal attire might indicate that this event is significant to him, or that he is in a moment of contemplation about the activity he is engaged in.

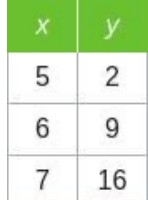

Question: <image>\nThe table shows a function. Is the function linear or nonlinear?

Answer: To determine whether the function is linear or nonlinear, see whether it has a constant rate of change.\nPick the points in any two rows of the table and calculate the rate of change between them. The first two rows are a good place to start.\nCall the values in the first row x1 and y1. Call the values in the second row x2 and y2.\nRate of change = \\frac{y2 - y1}{x2 - x1}\n = \\frac{9 - 2}{6 - 5}\n = \\frac{7}{1}\n = 7\nNow pick any other two rows and calculate the rate of change between them.\nCall the values in the second row x1 and y1. Call the values in the third row x2 and y2.\nRate of change = \\frac{y2 - y1}{x2 - x1}\n = \\frac{16 - 9}{7 - 6}\n = \\frac{7}{1}\n = 7\nThe two rates of change are the same.\nIf you checked the rate of change between rows 1 and 3, you would find that it is also 7.\nThis means the rate of change is the same for each pair of points. So, the function has a constant rate of change.\nThe function is linear.

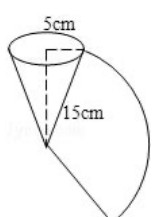

Question: <image>\nElaborate on what is shown in the illustration.

Answer: The diagram illustrates the lateral surface of a cone-shaped paper cup. The cone has a bottom surface, and the generatrix shows the slant height of the cone.

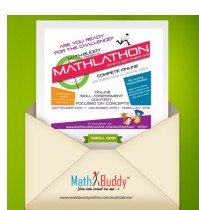

Question: <image>\nConsidering the detailed information provided about the event, what could be the potential benefits for a student to participate in this Mathlathon, based on the features advertised on the flyer?

Answer: The potential benefits for a student participating in the Mathlathon, as advertised, would include the opportunity to compete in an online setting with peers globally, which could enhance their competitive spirit and global awareness. The detailed reports could help students identify their strengths and weaknesses in math concepts. The excitement of winning trophies and medals could serve as a motivation to excel. Moreover, the possibility of being published in the Mathathon Hall of Fame could provide recognition for their achievements and could be a significant accolade for their academic portfolio.

## Novelty and Unseen Concepts

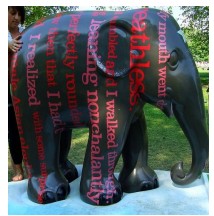

Question: <image>\nHow would you rate the quality of the photograph based on the provided information?

Answer: The photo looks very great, according to the image information.

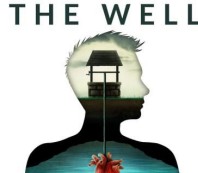

Question: What could be the symbolism of the man's silhouette with a heart hanging from a rope?\n<image>\n

Answer: The symbolism of a man's silhouette with a heart hanging from a rope could represent the concept of vulnerability and the emotional weight one might carry. The heart, often used as a symbol of love and emotions, is dangling from a rope, suggesting that it is exposed and perhaps fragile. The silhouette of the man might indicate that this is a universal experience, not limited to a specific individual. Overall, this image could symbolize the emotional struggles, vulnerability, and the challenges that people face in their inner lives.

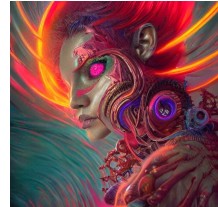

Question: <image>\nCould you tell me more about the artistic style used?

Answer: he artistic style is reminiscent of digital painting with vibrant colors and dynamic strokes, giving it a sense of motion. The details in both the mechanical and organic elements indicate a careful balance between realism and fantastical design.

Question: How might this image be interpreted symbolically?

Answer: Symbolically, the image could represent the intersection of humanity's natural essence with the advancement of technology. It might be expressing a perspective on the potential harmony or conflict between our biological origins and artificial enhancements.

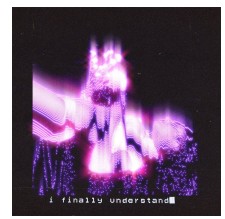

Question: <image>\nConsidering the variety of activities and the presence of diverse architecture and signage, what can be inferred about the cultural and economic characteristics of this city?

Answer: he image suggests a city with a rich cultural tapestry and a bustling economy. The diverse architectural styles indicate a blend of traditional and modern influences, possibly suggesting a city with a long history and a willingness to embrace contemporary trends. The presence of street vendors and a variety of shops signifies a thriving local economy with a strong focus on commerce and trade. The different activities of the people, from leisurely walks to engaging in commerce, imply a city with diverse socio-economic activities, catering to both locals and visitors. The intricate signage may also indicate a city that values artistic expression and attention to detail, further emphasizing its cultural richness.

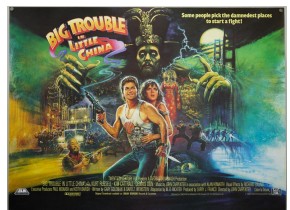

Question: <image>\nConsidering the stylistic elements and depicted characters, what genre does this poster suggest the movie belongs to, and what themes or motifs are suggested by the imagery used?

Answer: The poster suggests that the movie belongs to the action-adventure genre with elements of fantasy. The central figure with the gun, the action poses of the characters, and the ominous overlord figure with supernatural glowing eyes indicate a story that involves conflict, heroism, and otherworldly forces. The neon text, lightning effect, and the mystical object held by the seated older man imply themes of magic or supernatural power. The imagery of the street scene with varied characters in dynamic poses suggests motifs of a journey or quest, possibly with a group of diverse companions.

