# OpenReview forum: "ZooProbe: A Data Engine for Evaluating, Exploring, and Evolving Large-scale Training Data for Multimodal LLMs"
_ICLR.cc/2025/Conference — ICLR 2025 Poster_

### Official Review · Reviewer_p5bJ · 2024-10-27

**Soundness:** 3
**Presentation:** 3
**Contribution:** 3
**Rating:** 6
**Confidence:** 3

**Summary:**

This paper introduces ZooProbe - an innovative multimodal large language model (MLLM) training data engine. ZooProbe implements an Evaluate-Explore-Evolve (E3) cycle system that leverages a small-scale model zoo to evaluate data quality, explore high-quality data patterns, and strategically expand training datasets. The key innovations include: 1) proposing a 50+ dimensional data description vector system covering both intrinsic (e.g., resolution, language) and meta (e.g., annotation reliability, knowledge dependency) features; 2) developing an A*-search-based framework to optimize data distribution; 3) generating new data through rule-guided approaches. Experiments demonstrate that ZooProbe significantly outperforms traditional data scaling rules across scales from 10K to 260K examples.

**Strengths:**

1. The paper makes a clear technical contribution through its E3 cycle framework and model zoo approach. The innovation lies in systematically addressing data quality through a small-scale evaluation system, rather than relying on traditional scaling methods. The integration of A*-search for distribution optimization is technically sound and well-motivated.

2. The empirical evaluation is thorough and convincing. The authors provide comprehensive ablation studies across different scales, with the MLLM-Scorer showing consistent improvements over baselines. The experimental design effectively validates each component of the proposed system.

3. The work addresses a significant problem in MLLM training with practical implications. By reducing computational costs and carbon emissions while improving model performance, the approach offers valuable insights for the field. The methodology generalizes well to other domains, suggesting broader impact beyond the immediate application.

**Weaknesses:**

1. While the paper demonstrates improvements in data efficiency, the experimental evaluation lacks comparison with recent data-centric approaches such as ActiveLLM and DataSpell. These methods also address data quality optimization but through different mechanisms. Including these comparisons would better position the work's contributions within the current landscape.

2. The scalability of the approach beyond 260K examples remains unclear. While the results are promising at the demonstrated scales, larger industrial applications often deal with datasets in the millions. The authors should either provide theoretical guarantees for scalability or acknowledge this limitation explicitly. Additionally, the sensitivity of the method to the initial model zoo composition deserves more investigation - how much does performance vary with different zoo configurations?

**Questions:**

1. Regarding the scalability of ZooProbe:

    a. Have you tested or analyzed the approach's feasibility on datasets larger than 260K examples?

    b. Does the A* search optimization remain tractable for much larger datasets?

    c. Are there any theoretical bounds on the computational complexity as the dataset size grows?

2. Implementation details:

    a. Could you provide more details about the stopping criteria for the E3 cycle?

    b. How is the trade-off between exploration and exploitation managed in practice?

    c. What are the key hyperparameters that practitioners need to tune when applying ZooProbe?

---

> ### Author Response · Authors · 2024-11-24
>
> We sincerely appreciate Reviewer p5bJ for recognizing the practical significance of ZooProbe. Our responses are:
>
> * **Additional comparisons:** We further consider additional data synthesis and sampling methods, specifically evaluating ALLaVA [1] and ActiveLLM [2] results. However, DataSpell, as a data analysis IDE produced by JetBrains, may not correspond to a particular method. It facilitates data analysis by integrating SQL, Python, and other coding capabilities.
>
>   * ALLaVA introduces additional MLLM to generate instructions and annotations, and we sample it to the corresponding size for training.
>
>   * Similarly, we reimplement the *No Feedback Mode* of ActiveLLM. Given the restrictions of multi-image inference in current MLLMs, we employ Phi-3.5-mini-instruct for sample selection and Phi-3.5-vision-instruct as the annotation model.
>
>     | Performance on AI2D | 10k | 60k | 110k | 160k | 210k | 260k |
>     |-|-|-|-|-|-|-|
>     | LLaVA | 57.9 | 59.75 | 61.14 | 60.78 | 61.85 | 61.69 |
>     | ALLaVA | 57.9 | 59.55 | 61.76 | 61.33 | 60.98 | 62.14 |
>     | ActiveLLM | 57.9 | 58.84 | 61.01 | - | - | - |
>     | ZooProbe | 57.9 | 60.27 | 62.73 | 63.21 | 63.18 | 65.84 |
>
> * **Sensitivity of ZooProbe to the initial model zoo**: Firstly, we outline the evaluation dimensions and corresponding models (**details can be referred to** the response to Reviewer Xi4n), including:
>
>   * Intrinsic:
>     * basic dimensions (language, token length, brightness, color palette, resolution, visual clarity) that **do not** require additional models.
>     * Category-based: **GTE-Qwen2-Instruct** handles instruction & answer (textual) themes, while **CLIP** processes image (visual) themes.
>
>   * Fine-grained object-based evaluations are handled by **CLIP, SAM 2, and Depth Anything v2**, focusing on:
>     * visual foreground targets, number, type count, total area ratio, maximum area ratio, average distance from center
>     * visual background targets, type count
>     * maximum depth difference for different foreground targets, variance
>
>   - Meta dimensions processed by **GTE-Qwen2 Instruct, E5-V, and Phi-3-mini**, covering 28 multimodal and 10 textual dimensions.
>
>   In our response to Reviewer tve5, we present the results of **five additional ablation combinations**. The initial model zoo, which includes CLIP, SAM 2, and Depth Anything v2, is vital for capturing fine-grained semantics. Meanwhile, GTE-Qwen2 Instruct, E5-V, and Phi-3-mini significantly contribute to high-quality evaluations across meta dimensions, playing a crucial role in the continuous improvement of ZooProbe.
>
> * **Regarding the scalability of ZooProbe**:
>   * **Larger than 260K**: We scale up to **360k** (because we only have (8 + 4 + 4) $\times$ A100). We commit to scaling beyond 500K in the final version; it matches the data scale of LLaVA's SFT.
>
>   |360k|ai2d_test|mmmu_val|mmstar|realworldqa|scienceqa_test|seedbench_img|
>   |-|-|-|-|-|-|-|
>   |LLaVA|62.95|37.33|40.53|52.94|75.41|68.95|
>   |ZooProbe|64.54|38.69|42.8|53.33|76.45|69.51|
>
>   * **A* in much larger datasets**: The A* search algorithm is a crucial component of ZooProbe, specifically in the Exploring step of the E3 loop.
>     * The heuristic function in A* is derivable: the data subset sampled by Eq. 6 corresponds to a heuristic strategy $\pi$.
>     * Once $\pi$ is determined, the expansion process of A* involves continually sampling and evaluating new data, requiring only the inference process on the latest data. Consequently, the optimization step of A* remains effective in much larger datasets.
>   * **Computational bound**: we present this of the expansion process of ZooProbe:
>     1. Let the existing dataset be $D$, and each step of the expanded dataset be $d$. The training cost on dataset $d$ is $Tr(d)$, the inference cost is $Te(d)$, and the search window size is $N$.
>     2. Assume the size of the model zoo is $M$, and the cost of the evaluating step on the perturbation distribution for one searching window is $M \cdot N \cdot Te(d)$.
>     3. Obtaining the sampling weight for the target distribution in Eq. 6 requires $N \cdot Tr(D + d)$ (in this step, we can also optimize for efficiency with parameter-efficient fine-tuning or incremental training (Line 330)).
>     4. Extending $T$ steps with the chosen distribution $\pi$ necessitates an evaluation cost of $M \cdot Te(T \cdot d)$.
>     5. In conclusion, the construction cost is approximately:
>       $M \cdot N \cdot Te(d) + N \cdot Tr(D + d) + M \cdot Te(T \cdot d)$,
>
>       **which is roughly $\mathcal{O}(M \cdot Te(D))$ or $\mathcal{O}(N \cdot Tr(D))$.**
>
>     Since the model zoo consists of **small-scale (CV models) or embedding models** (allowing for batch inference), $M \cdot Te(D)$ is not large, and $M \cdot Te(T \cdot d)$ **can be mitigated** through peft and other incremental methods.
>
> **Thank you again for the reviewers' support.** We will continue to improve on ZooProbe, and **the implementation details are included in our subsequent response**.

---

> > ### Author Response · Authors · 2024-11-24
> > **Continuing the above response: implementation details**
> >
> > * **Implementation details**:
> >   * The E3 cycle is an automated data engine designed to scale based on the required magnitude. **It does not have explicit stopping criteria and can run until the dataset is sufficiently large.** It consists of three main components:
> >     1. It first **evaluates** various dimensions of the sample data.
> >     2. Based on this evaluation, it employs random perturbation (as noted in Line 320) to **explore** different distribution directions. Then, Eq. 5 is used to determine the expanded datasets corresponding to these varied directions.
> >     3. Finally, Eq. 6 is applied to calculate and sample a current strategy for expansion. If the sampled existing data does not align with the required strategies, we incorporate an MLLM to generate and evolve instructions and answers on corresponding images.
> >
> > * **Trade-off between exploration and exploitation**: In practice, the softmax parameter $\tau$ in Eq. 6 balances exploration and exploitation: a larger $\tau$ results in smoother sampling probabilities for more exploration, while a smaller $\tau$ favors the optimal policy $\pi$ for exploitation.
> >
> > * **Key hyperparameters**: The number of distribution perturbations, which corresponds to the size of the search gradient direction space (referred to as N), the size of the probe set, which corresponds to the search step size during data expansion, and the initial model zoo in Evaluating step. The number of update steps (referred to as T) is less critical, and can be 50, 20, or even 10.
> >
> > For other questions, please refer to the response above. Thank you.

---

> > > ### Author Response · Authors · 2024-12-01
> > >
> > > **We sincerely appreciate the time you have taken to review our paper.** In our response, we have provided additional implementation details and expanded our results. **Specifically**, we have clarified the dimensions evaluated (as addressed in our response to Reviewer Xi4n), detailed the dataset expansion process (to Reviewer 9FbC), conducted more extensive ablation studies (to Reviewer tve5), and explored scaling to a larger scale (to Reviewer p5bJ). **We extend our performance evaluations on** 410k, as detailed below:
> > >
> > > |410k|ai2d_test|mmmu_val|mmstar|realworldqa|scienceqa_test|seedbench_img|
> > > |-|-|-|-|-|-|-|
> > > |LLaVA (random sampling)|64.09|36.88|40.53|52.81|75.26|68.99|
> > > |LLaVA (ZooProbe)|65.38|38.46|43.33|53.46|76.85|69.44|
> > >
> > > Due to the brief nature of the author-reviewer discussion phase, **we would be grateful for your feedback on whether your primary concerns have been sufficiently addressed.** Please let us know if there are any points where you need further explanations or clarifications. **Thank you very much for your support and valuable insights!**

---

> > > > ### Comment · Reviewer_p5bJ · 2024-12-01
> > > >
> > > > Thank you for providing a detailed response to my concerns. While I acknowledge that you have addressed my technical questions and clarified the methodological aspects I raised, I will maintain my original overall assessment.
> > > >
> > > > Although your work is solid and the rebuttal has been informative, I believe the paper has not yet reached the higher bar required for an increased score. The current contribution, while valuable, still aligns with my initial evaluation in terms of its novelty and potential impact in the field.

---

> > > > > ### Author Response · Authors · 2024-12-01
> > > > >
> > > > > We deeply appreciate your support. We will continue to improve ZooProbe. Thank you.

---

### Official Review · Reviewer_tve5 · 2024-11-02

**Soundness:** 3
**Presentation:** 3
**Contribution:** 3
**Rating:** 5
**Confidence:** 3

**Summary:**

ZOOPROBE is a data engine designed to enhance the training of Multimodal Large Language Models (MLLMs) by evaluating, exploring, and evolving large-scale training data. The engine is built around an evaluating-exploring-evolving (E3) loop, which provides insights into data characteristics, identifies quality rules for data enhancement, and establishes a probe set for targeted rule selection tot facilitates the systematic evolution of new, high-quality data.

Experiments demonstrate that ZOOPROBE significantly beats scaling laws across scales from 10k to 260k

**Strengths:**

1. ZOOPROBE uses a small-scale model zoo consisting of various models to evaluate data comprehensively. These models assess over 50 dimensions of intrinsic and meta attributes, such as object and topic distribution, annotation quality, and scene complexity.
2. ZOOPROBE describes high-quality selection rules as an interpretable distribution based on the current data and model state. It uses Kullback-Leibler divergence to measure the difference in distribution and aims to achieve greater diversity in the dataset.
3. The authors have demonstrated through experiments and analysis of the results that ZOOPROBE is more effective and has lower overhead compared to other data augmentation strategies.

**Weaknesses:**

1. The paper presents ablation studies on the heuristic functions and evaluated dimensions, but it could benefit from more detailed analyses. For instance, it would be insightful to see how different combinations of intrinsic and meta dimensions impact the model's performance.

**Questions:**

1. Could the authors elaborate on the relative importance of the various intrinsic and meta dimensions in the data evaluation process? Are there any dimensions that consistently have a higher impact on model performance?

---

> ### Author Response · Authors · 2024-11-23
>
> Thank Reviewer tve5 for recognizing the zoo-based data evaluation in our ZooProbe. Our response is as follows:
>
> The main issue centers on "more ablations on different combinations of intrinsic and meta dimensions." **In Figure 4 (c)** of the main paper, we compare the impact of missing some evaluation dimensions on ZooProbe's performance and find that most dimensions have sufficient impact. Furthermore, we supplement more ablation combinations by **masking the influence of other dimensions and conduct experiments on the following types of dimension divisions respectively**:
>
> > Firstly, we **introduce all the dimensions** mentioned in Line 413 of the main paper.
> >
> > The intrinsic dimensions include:
> > * **Basic**: language, token length, brightness, color palette, resolution, visual clarity
> > * **Category-based**: instruction & answer (textual) theme, image (visual) theme
> > * **Fine-grained object-based**:
> >     * visual foreground targets, number, type count, total area ratio, maximum area ratio, average distance from center
> >     * visual background targets, type count
> >     * maximum depth difference for different foreground targets, variance
>
> > The meta dimensions include **28 multimodal-related** and **10 textual-related aspects**. We list 5 of each:
> > * **Multimodal**: Data Annotation Quality, Diversity of Visual Content, Cognitive Load, Accuracy of OCR Data, Mathematical Concept Coverage, and so on
> > * **Textual**: Linguistic, Conceptual, Knowledge, Ambiguity, Temporal Stability, and so on
>
> > A total of 6 + 2 + 10 + 28 + 10 = 56 dimensions. In the general response and reply to Reviewer Xi4n, we mention that each dimension corresponds to multiple assignments, such as each meta-dimension corresponding to high, medium, and low, and the visual object-related corresponding to 134 categories.
>
> |**Baseline & Ours**|60k|110k|160k|
> |-|-|-|-|
> |LLaVA|59.75|61.14|60.78|
> |All (ZooProbe on LLaVA)|59.88|62.73|63.96|
>
> **More ablation studies on AI2D dataset:**
>
> * Intrinsic dimensions **v.s.** Meta dimensions (Num of dims. 18 : 38).
>   |**Tab. 1**|60k|110k|160k|
>   |-|-|-|-|
>   |Intrinsic|59.94|63.15|62.31|
>   |Meta|59.78|62.31|62.56|
>
> * Intrinsic parts (6 : 2 : 10).
>   |**Tab. 2**|60k|110k|160k|
>   |-|-|-|-|
>   |Basic|58.94|60.52|60.72|
>   |Category-based|59.65|61.85|61.56|
>   |Fine-grained object-based|59.49|62.4|63.02|
>
> * Meta parts (28 : 10).
>   |**Tab. 3**|60k|110k|160k|
>   |-|-|-|-|
>   |Meta of Multimodal|59.68|62.11|62.76*|
>   |Meta of Textual|59.91|62.05|62.08|
>
> * Fine-grained object semantics (1 : 1 : 6).
>   |**Tab. 4**|60k|110k|160k|
>   |-|-|-|-|
>   |Instruction & answer (textual) theme|59.72|61.24|60.27|
>   |Image (visual) theme|59.59|62.37|61.72|
>   |Visual foreground|59.94|62.6|63.34|
>
> * Meta dimensions of *filtered* ones (Line 253 of the main paper) **v.s.** *scattered* ones (1 : 1 : 6).
>   |**Tab. 5**|60k|110k|160k|
>   |-|-|-|-|
>   |Image Quality & Data Annotation Quality (*filtered*)|59.65|60.88|61.46|
>   |Cognitive Load & Complexity of Visual Scenes (*scattered*)|60.14|62.53|61.27*|
>
>   where * means training on less than 160k for time reasons.
>
>
> **Analysis:**
>
> * In Tab. 1, we observe that ZooProbe on intrinsic dimensions enhances the scaling performance of LLaVA more effectively than meta ones **but lacks sufficient stability.** In contrast, meta dimensions demonstrate more robust improvements.
>
> * In Tab. 2, we explore **the impact of intrinsic dimensions.** Performance growth in the Basic category is likely insignificant because these features describe non-semantic information like length, resolution, and color palette. Category-based dimensions also show minimal performance boost, possibly due to the vast search space of our considered 5506 textual themes, and we validate it in Tab. 4. The fine-grained object-based dimension, however, significantly enhances performance, likely due to its precise semantic description of target objects.
>
> * In Tab. 3, regarding the meta dimensions, we find that both Multimodal and Textual-related features exhibit a stable performance growth trend, which is **positively related to** the results across all meta dimensions.
>
> * In Tab. 4, we further analyze classifications related to fine-grained semantics. Despite having only six dimensions, the Visual foreground target-based category contains more distinctive features **than other coarse-grained theme dimensions**, enhancing the better and more reliable training data. Additionally, the 'image (visual) theme' positively impacts performance, highlighting the importance of image diversity.
>
> * For the two types of meta dimensions mentioned in Line 253 of the main paper, as in Tab. 5, we find that the *filtered* dimensions can achieve good performance with a small amount of data, while *scattered* ones show a continuous performance improvement.
>
> **Thank you to Reviewer tve5 for their valuable suggestions.** In the final version, we will incorporate **comprehensive ablation experiments** and continue improving ZooProbe.

---

> > ### Author Response · Authors · 2024-12-01
> >
> > **We sincerely appreciate the time you have taken to review our paper.** In our response, we have provided additional implementation details and expanded our results. Specifically, we have clarified the dimensions evaluated (as addressed in our response to Reviewer Xi4n), detailed the dataset expansion process (to Reviewer 9FbC), conducted more extensive ablation studies (to Reviewer tve5), and explored scaling to larger scale (to Reviewer p5bJ).
> >
> > **Please feel free to reach out if you have further questions or concerns regarding the dimension ablation combinations, and we will do our utmost to address them.**
> >
> > Due to the brief nature of the author-reviewer discussion phase, we would be grateful for your feedback on whether your primary concerns have been sufficiently addressed. Please let us know if there are any points where you need further explanations or clarifications. **Thank you very much for your support and valuable insights!**

---

> > > ### Author Response · Authors · 2024-12-02
> > >
> > > As we approach **the FINAL day of the discussion period**, we kindly request Reviewer tve5 to **share any remaining concerns**. We are truly grateful for your insightful feedback.
> > >
> > > We hope that the additional experiments and analyses have addressed the earlier issues raised, and we would appreciate it if the final score could be **reconsidered**. Once again, **we extend our heartfelt thanks for your dedication and thoughtful input during the discussion period.**
> > >
> > > **Thank you!**

---

### Official Review · Reviewer_9FbC · 2024-11-03

**Soundness:** 3
**Presentation:** 2
**Contribution:** 2
**Rating:** 5
**Confidence:** 3

**Summary:**

This paper presents ZooProbe, a system for synthetic data generation and selection for multimodal tasks. ZooProbe is characterised by the evaluate-explore-evolve (E3) loop. Starting from some seed data, ZooProbe first assesses the quality of the data, and then employs A* search to iteratively grow the data. It also evolves the target data with rules.

Evaluation is conducted on a number of QA datasets including AI2D, ScienceQA, MMStar, RealWorldQA, SeedBench and MMMU. Results show that, in general, ZooProbe can elicit stronger performance than a random sample baseline, sometimes substantially.

**Strengths:**

* Synthetic data generation has become a key technique to continue to push VLM/LLM performance. Thus, investigations in this area are timely and important.

* The results show strong performance of the proposed technique.

**Weaknesses:**

* The clarity of the paper could be improved. Throughout the paper, there are phrases and sentences that are ambiguous (I will give some examples below). These ambiguities have an impact on the paper's readability.

* The impressive evaluation results are compared on a "random sample" baseline. Stronger data synthesis baselines should be compared.

**Questions:**

* The paper repeatedly claims that ZooProbe can "break the scaling law". It seems to mean ZooProbe is able to elicit strong performance from VLM. However, I'm not sure what that means exactly. Please clarify.

* Furthermore, this phrase ("breaking the scaling law") is hyperbolical, as it is only empirically validated on one single VLM (LLaVA) and six datasets. I suggest the authors tone it down.

* On line 107, the author mentions catastrophic forgetting. I agree that this is an important problem to be concerned with, and I'd like to see some discussions on how ZooProbe mitigates this forgetting problem.

---

> ### Author Response · Authors · 2024-11-23
>
> Thank Reviewer 9FbC for the recognition of our method and experiments. Our response is as follows:
>
> * **On "break the scaling law"**: In the final version, **we will tone down or remove these descriptions.** Instead, we will provide a more detailed explanation of the observed performance phenomenon. We intend to highlight that when using the training dataset filtered and generated by ZooProbe, MLLMs like LLaVA and Wings outperform traditional scaling laws for the given dataset size.
>   * Furthermore, **we expand to include more architecture**, *e.g.*, the performance of ZooProbe to LLaVA-Phi-3.5-mini-instruct:
>
>     | Performance on AI2D | 10k | 60k | 110k | 160k | 210k |
>     |-|-|-|-|-|-|
>     | LLaVA-Phi-3.5-mini-instruct | 63.05 | 64.83 | 66.87 | 67.65 | 68.23 |
>     | ZooProbe on it | 63.05 | 64.31 | 67.00 | 68.33 | 69.04 |
>
> * **Comparison of stronger data synthesis baselines**: We also compare ALLaVA [1] and ActiveLLM [2].
>   - ALLaVA introduces additional MLLM to generate instructions and annotations, and we sample it to the corresponding size for training.
>   - Similarly, we reimplement the *No Feedback Mode* of ActiveLLM. Given the restrictions of multi-image inference in current MLLMs, we employ Phi-3.5-mini-instruct for sample selection and Phi-3.5-vision-instruct as the annotation model.
>
>     | Performance on AI2D | 10k | 60k | 110k | 160k | 210k | 260k |
>     |-|-|-|-|-|-|-|
>     | LLaVA | 57.9 | 59.75 | 61.14 | 60.78 | 61.85 | 61.69 |
>     | ALLaVA | 57.9 | 59.55 | 61.76 | 61.33 | 60.98 | 62.14 |
>     | ActiveLLM | 57.9 | 58.84 | 61.01 | - | - | - |
>     | ZooProbe | 57.9 | 60.27 | 62.73 | 63.21 | 63.18 | 65.84 |
>
>     Resource constraints, particularly longer inference times with ActiveLLM, limited our evaluation to dataset sizes of 60k and 110k. ZooProbe's E3 loop can automatically and adaptively construct a dataset growth tree based on the model and existing data.
>
> - **Line 107 "catastrophic forgetting"**: The ZooProbe setting avoids incremental training. Here, $M^0$ is the model obtained after the alignment stage during MLLM training (that is, aligning the visual encoder and LLM using caption data). $M^1$ is trained on instruction data $D^1$ initialized with $M^0$. As the dataset grows, $D^2 = D^1 + d^1$. **To avoid incremental training,** $M^2$ is derived by performing instruction fine-tuning on $D^2$, starting with $M^0$, instead of fine-tuning incrementally on $d^1$ based on $M^1$.
> Similarly, $M^3$ is obtained by instruction fine-tuning on $D^3$, initialized again with $M^0$.
>
>     - Here, in ZooProbe's heuristic objective (Eq. 5 in the main paper), it calculates the change in the entire dataset distribution $d_{\text{diverse}} + \mathcal{D}^{\text{cur}}$ after adding a subset $d_{\text{diverse}}$, **rather than** focusing on the distribution of incremental part $d_{\text{diverse}}$.
>
>     - We compare the difference between instruction fine-tuning on the entire dataset initialized with $M^0$ *v.s.* incremental fine-tuning on $d$ initialized with $M^i$.
>
>       | Performance on AI2D | 10k | 60k | 110k |
>       |-|-|-|-|
>       | LLaVA (incre. from $M^i$) | 57.90 | 55.28 | 50.65 |
>       | LLaVA (tuning from $M^0$) | 57.9 | 59.75 | 61.14 |
>
> [**1**] ALLaVA: Harnessing GPT4V-synthesized Data for A Lite Vision-Language Model.
>
> [**2**] ActiveLLM: Large Language Model-based Active Learning for Textual Few-Shot Scenarios
>
> **Once again, we sincerely appreciate Reviewer 9FbC's support. We will incorporate all suggestions with sufficient experiments into the final version**. Thank you once again! **Please feel free to ask questions anytime.** We will continue to work hard to make ZooProbe better.

---

> > ### Comment · Reviewer_9FbC · 2024-11-27
> >
> > Thanks for your response.
> >
> > * In the 2 tables in your rebuttal, can you please elaborate what the baselines are? For instance, the row "LLaVA-Phi-3.5-mini-instruct" in the first table, does it represent this model fine-tuned with random data of 10k, 60k, etc.?
> >
> > * For catastrophic forgetting, there are specific metrics that measure the amount of forgetting. These include average accuracy, forward/backward transfer, etc. To really show ZooProbe effectively alleviates catastrophic forgetting, some empirical evidence should be provided.

---

> > > ### Author Response · Authors · 2024-11-29
> > >
> > > We really appreciate Reviewer 9FbC's feedback. Happy holidays!
> > >
> > > * **Q1:** In Table 1, **'LLaVA-Phi-3.5-mini-instruct'**, and in Table 2, **'LLaVA' are baseline models.**
> > >
> > >   And **yes, it represents this model fine-tuned with random data of 10k, 60k, etc.**
> > >
> > >   The LLaVA-Phi-3.5-mini-instruct is modified from LLaVA architecture by using Phi-3.5-mini-instruct as the LLM component and SigLIP as the visual encoder (a stronger MLLM backbone). MLLM training has two stages: caption alignment and instruction fine-tuning.
> > >
> > >   1. **Caption Alignment:** In the first stage, we align text-image features using LLaVA-1.5, ShareGPT4V, and ALLaVA-Caption datasets. We only train the connector that aligns the visual encoder with the LLM features (Line 404).
> > >
> > >   2. **Instruction Fine-tuning:** In the second stage, we fine-tune both the LLM and the connector using instruction data from the LLaVA-NeXT dataset. We randomly select 10k, then 60k (adding 50k more), 110k, and so on (Line 337).
> > >
> > > * **Q2:** We sincerely apologize for any confusion in our descriptions. **We do not claim that ZooProbe can alleviate forgetting, and our scaling settings do not lead to forgetting with incremental learning.** The confusion may come from line 107 of the main paper. There, we meant to say that while ZooProbe expands the dataset gradually, it does not involve incrementally training next model based on the previous one. Instead, we use the same model for training initialization.
> > >
> > >   **Next,** we will detail our ZooProbe setting to explain: (**1**) Why there is no incremental learning in ZooProbe's scaling law research; and (**2**) Although not explored in the main paper, we additionally supplement whether ZooProbe can alleviate model forgetting in a newly constructed setting:
> > >
> > > 1. ZooProbe's setting explores if higher-quality datasets can be obtained at various scales. We use **the same model** for initialization and fine-tune it on data of different sizes. Similar to studies on scaling laws, we compare model performance by training datasets of different sizes, like 10k, 60k, 110k, *i.e.*, **fine-tuning the model starting from the same initial state across these varying data scales.**
> > >
> > > 2. Line 107 tries to clarify this process: We start with an initialized MLLM, $M^0$, and **its performance on the 10k, 60k, and 110k scales is achieved by**:
> > >   * Fine-tuning $M^0$ on 10k, fine-tuning $M^0$ on 60k, and fine-tuning $M^0$ on 110k.
> > >   * **This differs from incremental training**, where $M^2$ is fine-tuned on a 50k increment with $M^1$ initialization, and then $M^3$ is further fine-tuned on another 50k increment with the $M^2$ initialization, and so on. Incremental training in this manner can lead to catastrophic forgetting.
> > >
> > >     More vividly,
> > >
> > >   * **Ours setting:** $M^0$ +10k -> $M^1$, $M^0$ +60k -> $M^2$, $M^0$ +110k -> $M^3$.
> > >
> > >   * **Incremental setting:** $M^0$ +10k -> $M^1$, $M^1$ +50k -> $M^2$, $M^2$ +50k -> $M^3$.
> > >
> > >   The previous response discusses how the second way, where each round of learning **occurs only on an incremental 50k** dataset, can lead to catastrophic forgetting.
> > >
> > > 3. In general encoder-based MLLM training, the second phase of instruction fine-tuning from $M^0$ (the visual feature is **aligned to** language semantic space) is not incremental learning because:
> > >    * The first phase mainly trains the connector **without the LLM part**.
> > >    * This phase aligns with visual descriptions, giving $M^0$ **weak instruction-following capability**.
> > >    * Incremental Learning (Task Incremental Learning (IL), Domain IL, or Class IL) **involves new task content**, unlike existing tasks. Instruction fine-tuning based on $M^0$ **includes image description learning**, so the learning content shows no incremental difference.
> > >
> > >    Therefore, although ZooProbe increases the dataset size step by step, from 10k to 60k, the model trains on the full 60k, **not just the incremental 50k.** We always start with the same model initialization with different data scales, so the **ZooProbe setting does not lead to incremental forgetting**.
> > >
> > > 4. While not covered in the main paper, we also construct a setting with incremental forgetting. We test if training data can help a model recover forgetting. We train $M^0$ using 100k math & Clevr [1] related instruction data. This new task focused on reasoning more, different from other multi-domain tasks, causing the model to forget other domain knowledge, and we compare the random *v.s.* ZooProbe on 100k data.
> > >
> > >     **Due to word limits, the table will be in the next response.** Our results show that ZooProbe improves the MLLM (LLaVA)'s overall performance, **even though it forgets** the knowledge in those domains.
> > >
> > > Thank you once again! Please feel free to share any questions or suggestions, and we will ensure that these distinctions are clearly explained in the final version. Thank you! We are committed to improving ZooProbe.
> > >
> > > [1] Clevr: A diagnostic dataset for compositional language and elementary visual reasoning.

---

> ### Author Response · Authors · 2024-11-29
> **Continuing the above response: results table**
>
> ||ai2d_test|mmmu_val|realworldqa|CCBench|MathVista_mini|
> |-|-|-|-|-|-|
> |LLaVA (10k)|57.9|**36.31**|51.11|30.78|29.6|
> |LLaVA (Train on Math & 'Clevr' related)|57.09|34.5|50.33|29.85|**31.9**|
> |Incre. FT on 100k based on Line 2 (Random)|59.88|36.2|50.2|31.18|31.2|
> |Incre. FT on 100k based on Line 2 (**ZooProbe**)|**60.98**|36.09|**52.23**|**31.32**|31.5|
>
> In this experiment, we build a setting where incremental learning on *math-related reasoning instructions* leads to forgetting knowledge in other domains. For more details, please see **the previous response**.

---

> > ### Author Response · Authors · 2024-12-01
> >
> > **We sincerely appreciate the time you have taken to review our paper.** In our response, we have provided additional implementation details and expanded our results. Specifically, we have clarified the dimensions evaluated (as addressed in our response to Reviewer Xi4n), detailed the dataset expansion process (to Reviewer 9FbC), conducted more extensive ablation studies (to Reviewer tve5), and explored scaling to larger scale (to Reviewer p5bJ).
> >
> > **Please feel free to reach out with any further questions or concerns regarding the ZooProbe data expansion setting. We are committed to resolving any issues to the best of our ability.**
> >
> > Due to the brief nature of the author-reviewer discussion phase, we would be grateful for your feedback on whether your primary concerns have been sufficiently addressed. Please let us know if there are any points where you need further explanations or clarifications. **Thank you very much for your support and valuable insights!**

---

> > > ### Author Response · Authors · 2024-12-02
> > >
> > > As we approach **the FINAL day of the discussion period**, we kindly request Reviewer 9FbC to **share any remaining concerns**. We are truly grateful for your insightful feedback.
> > >
> > > We hope that the additional experiments and analyses have addressed the earlier issues raised, and we would appreciate it if the final score could be **reconsidered**. Once again, **we extend our heartfelt thanks for your dedication and thoughtful input during the discussion period.**
> > >
> > > **Thank you!**

---

### Official Review · Reviewer_Xi4n · 2024-11-04

**Soundness:** 3
**Presentation:** 2
**Contribution:** 4
**Rating:** 6
**Confidence:** 3

**Summary:**

This paper presents a novel method to efficiently and effectively grow fine-tuning datasets of MLLMs. This method is built upon an evaluate-explore-evolve loop. In this loop, it utilizes other LLMs and smaller models to extract meta and intrinsic properties of additions to existing datasets, in order to estimate the performance of the base MLLM when trained on newer datasets and perform A* search based on this estimation. This method allows MLLMs to break scaling laws in multimodal instruction fine-tuning at scales of 260k, enabling MLLMs to systematically and automatically evolve training data in data/resource-constraint situations.

**Strengths:**

This paper presents a systematic approach towards a challenging problem–selecting and growing datasets to maximize downstream performance during fine-tuning. While this problem is often open-ended, and ill-formed, the proposed solution effectively utilizes and combines existing methods, by gathering signals using a zoo of powerful models and a well-defined search space for A* search. The implementation of such a system is also highly complex, incoporating many models and many dimensions of the data of interest. The significant primary and ablation results also represent that an impressive technical effort by the author(s) on the proposed method.

**Weaknesses:**

While this paper presents some impressive results on breaking the scaling law, I have a few concerns / questions about the paper. My first concern is around the possibility of overfitting to the evaluation benchmarks of interest. While the proposed method did not explicitly and directly optimize towards the evaluation benchmarks, I wonder if the benchmarks used to compute the reward overlaps with the evaluation. Even if they do not directly overlap, the set of benchmark would affect the generalizability of the proposed method to the general capabilities of the model. I believe this weakness can be resolved by having the author(s) further clarify the benchmarks used in reward/performance computation process in the loop. Another aspect of this weakness is whether duplication was observed between the candidate dataset and the ones used by the performance computation / evaluation benchmarks, of which an overlap of these images could indicate overfitting to particular images common in the benchmarks.

The second concern is the sparseness of details in the dimensions considered as meta and intrinsic properties of the data. While the author(s) have listed a number of examples of dimensions being considered, no specific/quantitative details were provided about these dimensions, and there is only a brief ablation on the number of dimensions in the results (which I appreciate). The paper can be improved with further exploration on the usefulness of various dimensions, and/or report failure cases on the impact of specific/specific types of dimensions on the effectiveness of the proposed method.

My final concern would be the strong assumption of targetting the uniform distribution (or random perturbations that starts from the uniform distribution) for the high-quality rules. The paper can be improved by further explaining the most common perturbations that the methods have converged to, and how future method(s) might consider a different distribution to begin with.

**Questions:**

Directly related to my concerns above:

- What are the benchmarks used for performance computation in the reward / heuristics?
- Are there duplication in the images used for the benchmarks used for reward computation and the evaluation benchmarks?
- Can you provide more details on the dimensions used (i.e., a detailed table / number of dimensions for meta/intrinsic properties of data)?
- Which dimensions/properties contributes most/least to high performance of the method?
- Can you provide more details about the perturbed distributions, such as which of them are commonly observed, and how much does the perturbation matter (compared to targetting uniform distribution)?

---

> ### Author Response · Authors · 2024-11-24
>
> We are very grateful to the Reviewer Xi4n for the support of ZooProbe. Our responses are:
>
> * **The benchmarks used for performance computation in the reward/heuristics** are derived from a sampled sub-dataset of evaluation benchmarks (as indicated in Line 299). We construct a validation set with about **1.5%** (15 * 5 + 65 * 1 + 6 * 10 + 1 * 50 + 9 * 5 + 30 * 2 = 355 samples) of the total evaluation benchmarks. As detailed below, we sample around 50 examples from each evaluation dataset, including AI2D, ScienceQA, MMStar, RealWorldQA, SEEDBench, and MMMU, *e.g.*, in order, 15 * 5 indicates that we sampled 5 from each of the 15 sub-class of AI2D. To further test cross-dataset capabilities, we evaluate additional benchmarks:
>
> |MME|10k|60k|110k|160k|210k|260k|
> |-|-|-|-|-|-|-|
> |LLaVA|1606.17|1566.30|1655.56|1667.69|1737.47|1698.96|
> |ZooProbe|1606.17|1698.09|1759.84|1768.77|1798.19|1794.98|
>
> |MathVista_mini|10k|60k|110k|160k|210k|260k|
> |-|-|-|-|-|-|-|
> |LLaVA|29.6|29.7|29.2|30.2|30.5|30.4|
> |ZooProbe|29.6|30|30.2|30|31.2|30.9|
>
> |CCBench|10k|60k|110k|160k|210k|260k|
> |-|-|-|-|-|-|-|
> |LLaVA|30.78|31.67|31.52|30.98|32.55|32.65|
> |ZooProbe|30.78|32.01|32.21|32.55|33.09|32.99|
>
> * **More details of dimensions:**
>
>   * **Intrinsic:**
>
> |Dimension|Num of Feature|Feature Description|
> |-|-|-|
> |language|4|AR, EN, RU, ZH|
> |token length|5|*|
> |brightness|2|dark, bright|
> |color palette|14|base colors|
> |resolution|3|^|
> |visual clarity|3|blurry, moderate, sharp|
> |sum|31|-|
> |-|-|-|
> |textual theme|5506|BISAC book classification|
> |image theme|134|SoM (Set-of-Mark)|
> |sum|5640|-|
> |-|-|-|
> |foreground targets|134|SoM|
> |fore. number|5|*|
> |fore. type count|5|*|
> |fore. total area ratio|3|^|
> |fore. max area ratio|3|^|
> |fore. average distance from center|3|^|
> |background targets|134|SoM|
> |back. type count|5|*|
> |max depth difference|3|^|
> |depth variance|3|^|
> |sum|298|-|
>
> where **symbol * and ^** indicate that the corresponding buckets (number 5 or 3) are divided according to numerical value from small to large. Despite the textual theme having the most features, Eq. 5 uses KL calculations for each dimension, normalizing the influence so all 56 dimensions **affect the result equally**.
>
>   * **Meta:**
>
> **Multimodal (28)**: Data Annotation Quality,Diversity of Visual Content,Object Recognition Difficulty,Image Quality,Complexity of Visual Scenes,Dependence on Visual Information,Dependence on Textual Information,Interplay between Questions and Images,Requirement for External Knowledge,Integration of Multiple Modalities,Temporal Sequences,Contextual Interpretation,Action Dynamics,Emotion Recognition,Multi-lingual Complexity,Detail Orientation,Accuracy of OCR Data,Table Structure Complexity,Cognitive Load,Novelty and Unseen Concepts,Mathematical Concept Coverage,Numerical Data Interpretation,Step-by-Step Problem Solving,Sarcasm and Irony Recognition,Pose and Gesture Interpretation,Spatial Reasoning,Domain-Specific Knowledge,Role of Graphical Elements
>
> **Textual (10)**: Linguistic,Conceptual,Answer Difficulty,Knowledge,Logic,Ambiguity,Speculation,Subjective,Temporal Stability,Practicality
>
> where the prompts like: 'Data Annotation Quality: Evaluate the quality and reliability of annotations provided, including labeling, bounding boxes, and metadata ..'
>
> * **Dimensions contributions:** Please refer to the response to Reviewer tve5, we conduct detailed ablations and find the meta dimensions slightly more robust than the intrinsic ones. In the intrinsic ones, fine-grained semantics outperform non-semantic features or coarse-grained themes. Image quality is a little more critical. In the meta ones, both multimodal and text-related aspects consistently enhance performance. The *filtered* approach effectively selects reliable samples with limited data, while the *scattered* type offers continual support.
>
> * **Perturbation distribution details**: In Line 431, we mention applying adequately random perturbations based on a uniform distribution. The perturbation one aligns with the overall dataset's evolving direction. For instance, if existing data distribution of two dimensions is `7:3`, and the target (perturbation one) is `9:1`, adding new dataset $d$ may shift the existing data distribution to `7.1:2.9`, thus moving closer to the target. Here, the perturbation target is focused on the relative direction, with Eq. 6 used to explore better directions.
>
>   * In the ZooProbe on two architectures, we initially categorize 134 image themes into head and tail classes. Categories like "person," which appear frequently, fall into the head class. We find that 70% of the adopted perturbation target distribution, which optimization direction tends towards increasing tail classes and decreasing head classes.
>
> - In Figure 4(b) of the main paper, we compare ZooProbe **with uniform distribution (Fixed Rule, red dashed line)**, with related analysis found in Line 512.
>
> **Thank you very much.** We will add comprehensive content in the final version and **continue improving**.

---

> > ### Comment · Reviewer_Xi4n · 2024-11-28
> > **Reviewer Response**
> >
> > Thank you for your detailed responses to my questions. Regarding these "additional benchmarks": Are they (MathVisa_mini, CCBench, MME) not included in the validation set used for performance computation?

---

> > > ### Author Response · Authors · 2024-11-29
> > >
> > > **Yes**, 'MathVisa_mini, CCBench, MME' are not included in the validation set.
> > >
> > > Thank you very much for your support. We will incorporate your suggestions into the final version. Have a great holiday!

---

> > > > ### Author Response · Authors · 2024-12-01
> > > >
> > > > **We sincerely appreciate the time you have taken to review our paper.** In our response, we have provided additional implementation details and expanded our results. Specifically, we have clarified the dimensions evaluated (as addressed in our response to Reviewer Xi4n), detailed the dataset expansion process (to Reviewer 9FbC), conducted more extensive ablation studies (to Reviewer tve5), and explored scaling to larger scale (to Reviewer p5bJ).
> > > >
> > > > **Should you have any further questions regarding the data evaluation dimensions, please do not hesitate to reach out.** We are committed to addressing any concerns you may have to the best of our ability.
> > > >
> > > > Due to the brief nature of the author-reviewer discussion phase, we would be grateful for your feedback on whether your primary concerns have been sufficiently addressed. Please let us know if there are any points where you need further explanations or clarifications. **Thank you very much for your support and valuable insights!**

---

> > > > > ### Author Response · Authors · 2024-12-02
> > > > >
> > > > > As we approach **the FINAL day of the discussion period**, we kindly request Reviewer Xi4n to **share any remaining concerns**. We are truly grateful for your insightful feedback.
> > > > >
> > > > > We hope that the additional experiments and analyses have addressed the earlier issues raised, and we would appreciate it if the final score could be **reconsidered**. Once again, **we extend our heartfelt thanks for your dedication and thoughtful input during the discussion period.**
> > > > >
> > > > > **Thank you!**

---

### Meta-Review · Area_Chair_Vsq4 · 2024-12-23

**Metareview:**

The paper presents a new method to incrementally grow fine-tuning datasets of MLLMs. The framework uses small-scale models to "evaluate" data quality using small scale models, "explore" high quality data that should be added to the dataset, and "evolve" the training data to cover gaps in the existing dataset. The authors show that the proposed data engine beat random sampling scaling laws using open-source VLMs like LLAVA.

All reviewers agreed that the paper makes a solid technical contribution, and show strong performance using the proposed techniques. Authors also addressed major concerns raised by reviewers about testing the technique to larger dataset size, comparison to other data synthesis baseline, and perform detailed analysis and discussion on how different combination of intrinsic and meta dimensions impact model's performance. Building efficient data pipelines has become an important area of research in recent times, and the contributions of the work will be useful and timely to that community.

**Additional Comments On Reviewer Discussion:**

Improvements in writing: Multiple reviewers pointed out that writing can be improved. We recommend the authors to proofread the paper more closely and incorporate all suggestions made during the discussion period in the final manuscript.

- Toning down the "break the scaling law" message. As promised by the authors, we expect that they will tone down these descriptions and make a more precise statement about the observations in the paper.

- Clarifications about the catastrophic forgetting. Authors and reviewer 9fBc had back-and-forth clarification dialog around catastrophic forgetting effect of the proposed approach. This could be a common point of confusion and we recommend the authors to incorporate these clarifications in the paper.

- Clarifications about scalability: As the authors promised, showing scalability upto the size of current SFT data will be greatly appreciated. Additionally, additional experiments conducted during the discussion period should be added to the appendix or the main manuscript.

---

### Decision · Program_Chairs · 2025-01-22

Accept (Poster)